# *Mir204* and *Mir211* suppress synovial inflammation and proliferation in rheumatoid arthritis by targeting *Ssrp1*

Qi-Shan Wang[1†], Kai-Jian Fan[1†], Hui Teng[1], Sijia Chen[1], Bing-Xin Xu[1], Di Chen[2]*, Ting-Yu Wang[1]*

[1]Department of Pharmacy, Shanghai Ninth People's Hospital, Shanghai Jiao Tong University School of Medicine, Shanghai, China; [2]Faculty of Pharmaceutical Sciences, Shenzhen Institute of Advanced Technology, Chinese Academy of Sciences, Shenzhen, China

**\*For correspondence:**
drtywang@163.com (T-YuW);
di.chen@siat.ac.cn (DC)

[†]These authors contributed equally to this work

**Abstract** Rheumatoid arthritis (RA) is a chronic inflammatory joint disease characterized by synovial hyperplasia. *Mir204* and *Mir211* are homologous miRNAs with the same gene targeting spectrum. It is known that *Mir204/211* play an important role in protecting osteoarthritis development; however, the roles of *Mir204/211* in RA disease have not been determined. In the present study, we investigated the effects and molecular mechanisms of *Mir204/211* on synovial inflammation and hyperproliferation in RA. The effects of *Mir204/211* on the inflammation and abnormal proliferation in primary fibroblast-like synoviocytes (FLSs) were examined by *Mir204/211* gain-of-function and loss-of-function approaches in vitro and in vivo. We identified the structure-specific recognition protein 1 (*Ssrp1*) as a downstream target gene of *Mir204/211* based on the bioinformatics analysis. We overexpressed *Ssrp1* and *Mir204/211* in FLS to determine the relationship between *Ssrp1* and *Mir204/211* and their effects on synovial hyperplasia. We created a collagen-induced arthritis (CIA) model in wild-type as well as *Mir204/211* double knockout (dKO) mice to induce RA phenotype and administered adeno-associated virus (AAV)-mediated *Ssrp1-shRNA* (AAV-*shSsrp1*) by intra-articular injection into *Mir204/211* dKO mice. We found that *Mir204/211* attenuated excessive cell proliferation and synovial inflammation in RA. *Ssrp1* was the downstream target gene of *Mir204/211*. *Mir204/211* affected synovial proliferation and decelerated RA progression by targeting *Ssrp1*. CIA mice with *Mir204/211* deficiency displayed enhanced synovial hyperplasia and inflammation. RA phenotypes observed in *Mir204/211* deficient mice were significantly ameliorated by intra-articular delivery of AAV-*shSsrp1*, confirming the involvement of *Mir204/211-Ssrp1* signaling during RA development. In this study, we demonstrated that *Mir204/211* antagonize synovial hyperplasia and inflammation in RA by regulation of *Ssrp1*. *Mir204/211* may serve as novel agents to treat RA disease.

## Editor's evaluation

This important study provides new understanding on the role of miR-204/-211 in the progression of rheumatoid arthritis and the underlying mechanisms. The evidence supporting the conclusions is convincing, with rigorous in vitro cell culture assays and in vivo mouse studies. The work will be of interest for skeletal biologists studying the pathogenesis of rheumatoid arthritis.

## Introduction

Rheumatoid arthritis (RA) is a chronic systemic disease, mainly characterized by synovitis, synovial hyperplasia, pannus formation, and osteochondral destruction (*Muraki et al., 2018*; *Shiraishi et al.,*

2017). Due to the joint dysfunction caused by RA, patients often have limited activities, and the quality of life is seriously reduced (*Wang et al., 2020a*). Although many effective drugs have been developed, the long treatment cycle of RA significantly increases the incidence of adverse drug reactions. Therefore, to provide the basis for individualized therapy, it is vital to elicit the molecular mechanisms of RA pathogenesis.

MicroRNAs (*miRNAs*) are small, non-coding RNA molecules consisting of 21–24 base pairs that specifically bind to target genes through the 3' untranslated region (3'-UTR) and control the expression of multiple genes at the post-transcriptional level (*Zhang et al., 2019*). In recent years, cumulative evidences have shown that *miRNAs* play crucial roles in the development of RA by regulating cell viability, apoptosis, and invasion (*Lai et al., 2017*; *Stanczyk et al., 2008*). In synovial tissue of patients with RA, *miRNA* expression was significantly altered compared with those of healthy individuals. The abnormal *miRNAs* promote the expression of proinflammatory cytokine and enzymes that erodes the cartilage matrix by interfering Wnt, NF-κB, and JAK/STAT signaling pathways (*Han et al., 2022*).

The *Mir204* and *Mir211* have similar nucleotide sequences and share the same seed sequence. There are only two nucleotide differences in humans and one nucleotide difference in mice. These structural similarity and homology between these two *miRNAs* allow them to have the same gene targeting spectrum (*Lee et al., 2016*). In breast cancer, *Mir204* acts as a tumor suppressor. The expression of *Mir204* induced apoptosis while the deletion of *Mir204* gene led to abnormal cell invasion (*Imam et al., 2012*). *MiRNA* has also become an indispensable regulatory factor in bone pathophysiology (*Asahara, 2016*). We have previously reported that homologous *miRNAs*, *Mir204* and *Mir211*, together protecting joint from osteoarthritis development. The loss of *Mir204/211* led to the upregulation of matrix degradation enzymes in articular chondrocytes, resulting in articular cartilage destruction and synovial hyperplasia (*Huang et al., 2019*). Since synovial inflammation is now considered as the major pathology of RA and hyperplasia of fibroblast-like synoviocyte (FLS) displayed tumor-like behaviors that contribute to pannus growth, inflammation, and cartilage damage, here we aimed to investigate whether *Mir204/211* play an indispensable role in RA synovial inflammation and hyperplasia (*Winchester et al., 2015*).

Structure-specific recognition protein 1 (SSRP1) is a subunit of the histone chaperone transcription (FACT) complex and is involved in almost all chromatin-related processes. As a transcription factor, *Ssrp1* regulates multiple cell processes, including cell cycle regulation and DNA repair (*Gao and Xiong, 2018*; *Wang et al., 2019*). In addition, *Ssrp1* expression was significantly upregulated in a variety of tumors, such as breast and ovarian cancer, and was strongly associated with poorer prognosis. The knockdown of *Ssrp1* expression by *siRNA* technique depressed the proliferation of glioma cells (*Liao et al., 2017*). SSRP1 inhibitor, CBL0137, induced cell death by down-regulating gene expression in the NF-κB signaling pathway (*Barone et al., 2017*). Wu et al. suggested that *Ssrp1* modulates P53 pathway and its downstream molecules, and *Ssrp1* may be involved in the cell cycle progression through regulation of P53 pathway (*Wu et al., 2019a*). Although *Ssrp1* has been shown to stimulate proliferation through cell cycle regulation in a variety of human cancers, the role, mechanism, and clinical significance of *Ssrp1* in RA remain unclear.

Collagen-induced arthritis (CIA) shares similar characteristics with RA, including synovial hyperplasia and inflammatory articular cartilage destruction, and has been extensively used in RA research (*Song et al., 2018*; *Bas et al., 2016*). Multiple lines of evidence have shown that RA FLSs are the principal cells in participation in the initiation, development, and deterioration of joint inflammation in RA disease (*Xiao et al., 2021*). In this study, we determined the effects of *Mir204/211* on the inflammatory response and proliferation of primary CIA FLS by overexpression or knockdown of *Mir204/211*, as well as the downstream mechanisms of *Mir204/211* in mitigation of synovial inflammation and excessive proliferation. We determined the relationship between *Mir204/211* and *Ssrp1* and revealed their effects on CIA-induced synovial cell proliferation in *Mir204/211* double knockout (dKO) mice. Our findings suggest that *Mir204/211* suppressed synovial inflammation and proliferation in RA by targeting *Ssrp1*.

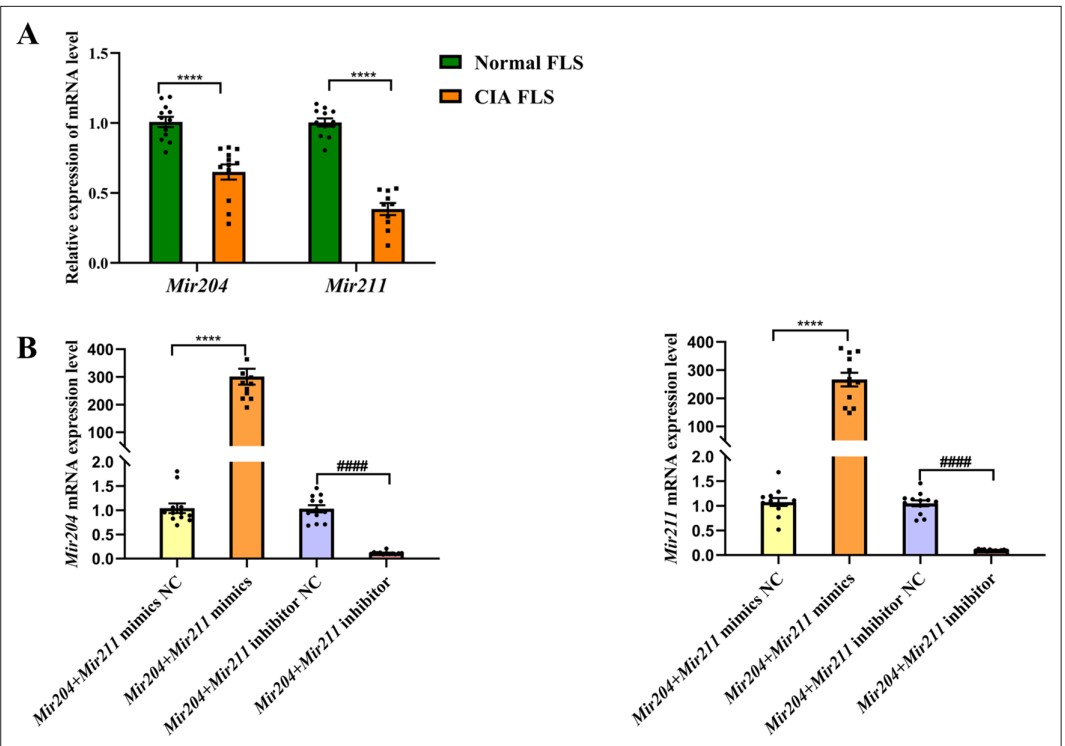

**Figure 1.** *Mir204/Mir211* downregulate in fibroblast-like synoviocyte (FLS) of collagen-induced arthritis (CIA) mice. (**A**) Low expression levels of *Mir204/Mir211* in FLS of CIA mice. (**B**) qRT-PCR analysis of the relative *Mir204/Mir211* expression in CIA FLS 48 hr after transfection. Please see *Figure 1—source data 1*. Data are pooled from at least three independent experiments and are presented as mean ± SEM. Data are analyzed using unpaired two-tailed Student t test (**A**) and one-way ANOVA (**B**). $^*p<0.05$, $^{**}p<0.01$, $^{***}p<0.001$, $^{****}p<0.0001$; and $^\#p<0.05$; $^{\#\#}p<0.01$, $^{\#\#\#}p<0.001$, $^{\#\#\#\#}p<0.0001$.

The online version of this article includes the following source data and figure supplement(s) for figure 1:

**Source data 1.** Numerical data obtained during experiments represent in *Figure 1*.

**Figure supplement 1.** Transfection efficiency of *Mir204/Mir211* in CIA FLS and viability of CIA FLS with different treatments.

## Results

### *Mir204/Mir211* downregulate in FLS of CIA mice

Using FLS of normal mice as the control group and U6 as the internal reference gene, we detected *Mir204/Mir211* expression in FLS of CIA mice (*Figure 1—figure supplement 1*). As seen in *Figure 1A*, expression levels of *Mir204* and *Mir211* were significantly lower in FLS of CIA mice than those of normal mice (p<0.0001). These findings suggest that *Mir204/Mir211* may play an important role in RA development.

According to PCR results in *Figure 1B*, the expression levels of *Mir204* and *Mir211* in CIA FLS transfected with two types of mimics were significantly higher compared with the NC mimics transfected group. Compared with the NC inhibitor group, the expression levels of *Mir204* and *Mir211* in CIA FLS transfected with both inhibitors were significantly decreased.

### *Mir204/Mir211* affect apoptosis and cell migration ability of CIA FLS

As shown in *Figure 2A*, compared with the NC mimics group, the overexpression of *Mir204/Mir211* promoted the apoptosis of CIA FLS whereas knockdown of *Mir204/211* inhibited the apoptosis rates of CIA FLS in comparison with the NC inhibitor group, suggesting that *Mir204/Mir211* are potent regulators of FLS apoptosis.

As it can be seen from *Figure 2B*, there was no marked difference among the four groups of transfected CIA FLS at 0 hr. However, at 24 hr, co-transfection of *Mir204/Mir211* markedly restrained the

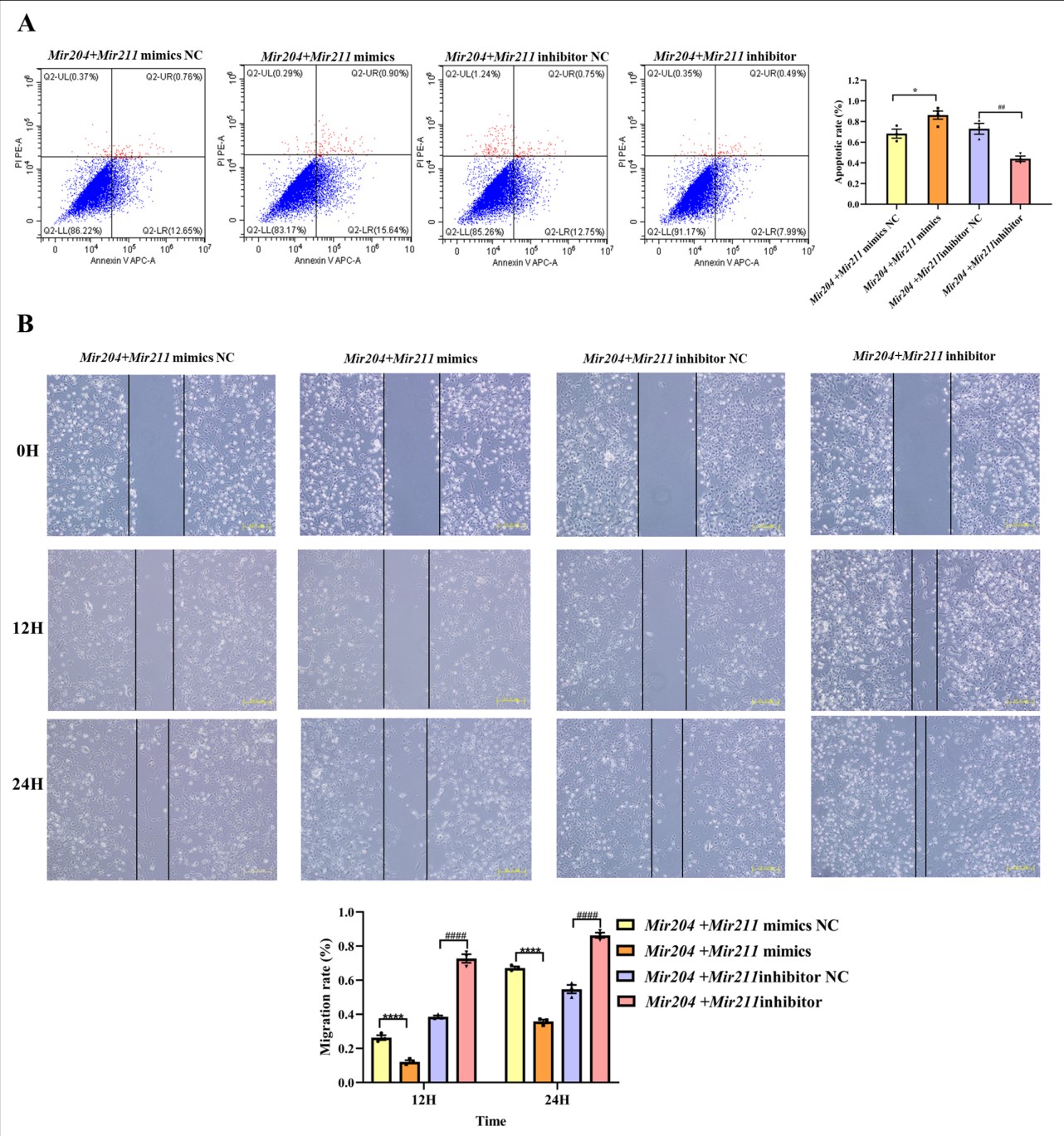

**Figure 2.** Effects of *Mir204/Mir211* on apoptosis and cell migration ability of collagen-induced arthritis (CIA) fibroblast-like synoviocyte (FLS). (**A**) Effects of *Mir204/Mir211* on apoptosis of CIA FLS. (**B**) Effects of *Mir204/Mir211* on cell migration ability of CIA FLS. Representative photomicrographs show a wound scratch assay at specific time points (0, 12, and 24 hr). Please see *Figure 2—source data 1*. Data are pooled from three independent experiments and are presented as mean ± SEM. Data are analyzed using one-way ANOVA. $^*p<0.05$, $^{**}p<0.01$, $^{***}p<0.001$, $^{****}p<0.0001$; and $^\#p<0.05$; $^{\#\#}p<0.01$, $^{\#\#\#}p<0.001$, $^{\#\#\#\#}p<0.0001$.

The online version of this article includes the following source data for figure 2:

**Source data 1.** Numerical data obtained during experiments represent in *Figure 2*.

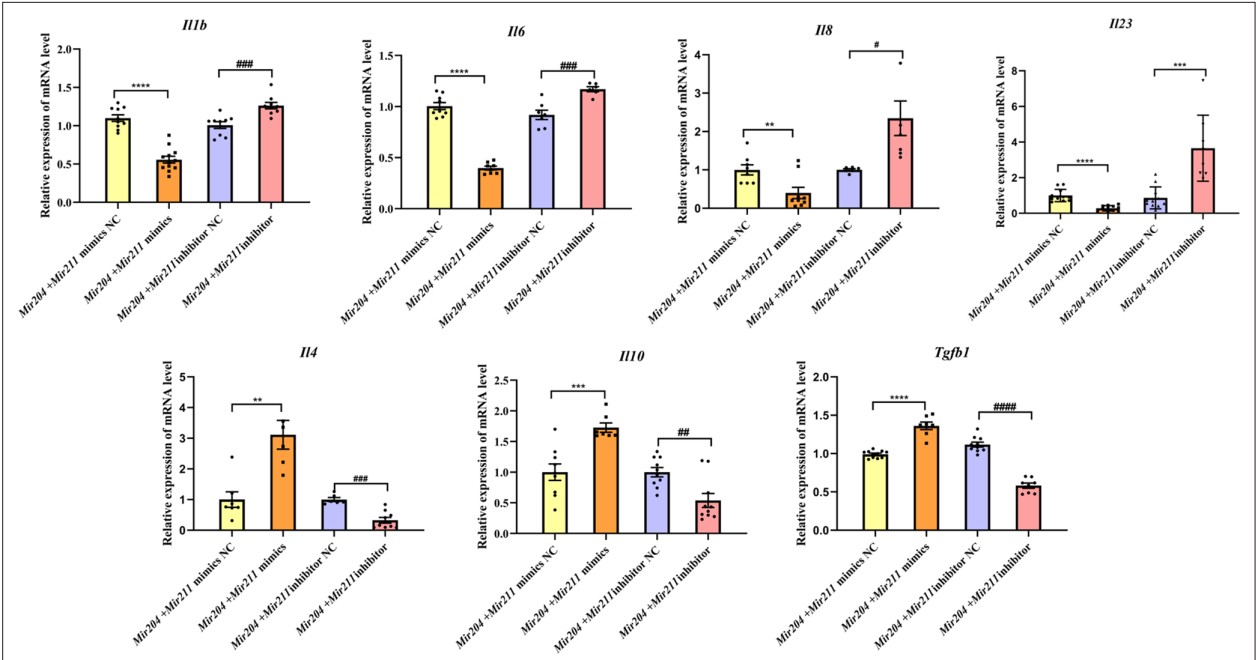

**Figure 3.** Effects of *Mir204/Mir211* on the inflammatory responses of collagen-induced arthritis fibroblast-like synoviocyte. All data are displayed as a value relative to those in the NC group. Please see *Figure 3—source data 1*. Data are pooled from at least three independent experiments and are presented as mean ± SEM. Data are analyzed using one-way ANOVA. *p<0.05, **p<0.01, ***p<0.001, ****p<0.0001; and #p<0.05; ##p<0.01, ###p<0.001, ####p<0.0001.

The online version of this article includes the following source data for figure 3:

**Source data 1.** Numerical data obtained during experiments represent in *Figure 3*.

migration of CIA FLS in comparison with the NC mimics group (p<0.0001). Compared with the NC inhibitor group, knockdown of both *Mir204/Mir211* greatly enhanced the migration ability of CIA FLS (p<0.0001). These results indicated that *Mir204/Mir211* also play an important role in regulation of FLS migration.

### *Mir204/Mir211* ameliorate the inflammatory responses of CIA FLS

Both proinflammatory and anti-inflammatory cytokines play a vital role in synovial inflammation in RA. Therefore, the expression levels of pro-inflammatory mediators (*Il1b*, *Il6*, *Il8*, and *Il23*) and anti-inflammatory factors (*Il4*, *Il10*, and *Tgfb1*) were detected by qRT-PCR. As shown in *Figure 3*, the mRNA levels of *Il1b*, *Il6*, *Il8*, and *Il23* were all decreased in FLS co-transfected with *Mir204/Mir211* mimics, and mRNA levels of *Il4*, *Il10*, and *Tgfb1* were all increased compared with FLS co-transfected with the *Mir204/Mir211* mimics NC. Compared with FLS transfected with *Mir204/Mir211* inhibitor NC, expression of pro-inflammatory cytokines was up-regulated, and expression of anti-inflammatory cytokines was down-regulated in FLS transfected with *Mir204/Mir211* inhibitors. The data above corroborated the effects of *Mir204/Mir211* on reducing synovial inflammation by down-regulation of pro-inflammatory cytokines and up-regulation of anti-inflammatory factors.

### *Mir204/Mir211* influence synovial inflammation by regulating NF-κB signaling pathway and p65 nuclear translocation

To further elucidate that *Mir204/Mir211* attenuate the inflammation of CIA FLS, we performed western blot analysis of key molecules in the NF-κB signaling. The results showed that compared with FLS transfected with the NC mimics, expression of NF-κB p65 and IKK-α was decreased in CIA FLS co-transfected with *miR-204/211*, while expression of I-κBα protein was increased (*Figure 4A*). Compared with the FLS transfected with NC inhibitors, knockdown of both *Mir204/Mir211* increased the protein levels of NF-κB p65 and IKK-α and decreased I-κBα protein levels. However, neither overexpression nor knockdown of *Mir204/Mir211* had significant effects on IKK-β expression. These

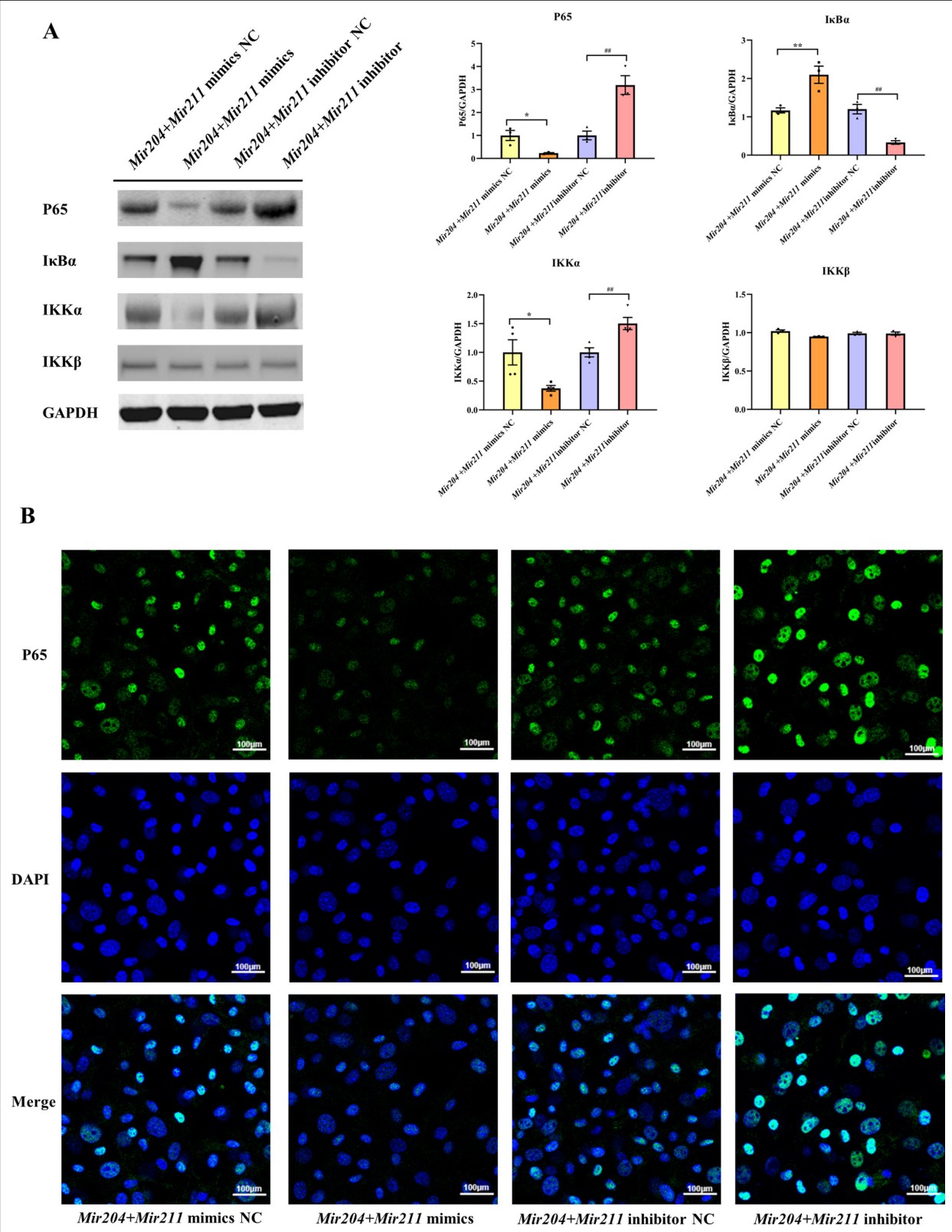

**Figure 4.** Effects of *Mir204/Mir211* on NF-κB signaling pathway. (**A**) Western blot assays of the NF-κB pathway in transfected collagen-induced arthritis (CIA) fibroblast-like synoviocyte (FLS). (**B**) Immunofluorescence staining of p65 nuclear translocation in transfected CIA FLS. Please see *Figure 4—source data 1*. Data are pooled from at least three independent experiments and are presented as mean ± SEM. Data are analyzed using one-way ANOVA. *p<0.05, **p<0.01, ***p<0.001, ****p<0.0001; and #p<0.05, ##p<0.01, ###p<0.001, ####p<0.0001.

*Figure 4 continued on next page*

*Figure 4 continued*

The online version of this article includes the following source data for figure 4:

**Source data 1.** Numerical data obtained during experiments represent in *Figure 4*.

**Source data 2.** Original western blot files for *Figure 4*.

results suggest that *Mir204/Mir211* might affect the CIA synovial inflammation by regulation of NF-κB signaling pathway.

We also performed immunofluorescence (IF) staining to detect the nuclear translocation of p65 in transfected CIA FLS. As shown in *Figure 4B*, compared with the FLS transfected with NC mimics, the nuclear translocation of p65 in CIA FLS co-transfection with both *Mir204* and *Mir211* was inhibited. In contrast, compared with the FLS transfected with NC inhibitors, p65 nuclear translocation was significantly enhanced in CIA FLS with both *Mir204* and *Mir211* knockdown, indicating that *Mir204/Mir211* inhibit NF-κB signaling by blocking p65 nuclear translocation.

### *Mir204/Mir211* influence proliferation of CIA FLS

*Mir204* mimics and *Mir211* mimics were co-transfected into CIA FLS, and the viability of CIA FLS was significantly decreased (*Figure 5A*, p<0.001). After transfection with *Mir204* inhibitor and *Mir211* inhibitor, the viability of CIA FLS was significantly increased (p<0.0001), suggesting that *Mir204/Mir211* are important regulators of FLS proliferation during RA development. Staining of β-galactosidase was also performed in primary WT FLS and dKO FLS (*Figure 5—figure supplement 1*).

We also examined effects of *Mir204/Mir211* on cell cycle progression in CIA FLS using flow cytometry and RT-qPCR assays. Compared with the NC mimics group, overexpression of both *Mir204* and *Mir211* increased the proportion of G0/G1 phase and reduced the proportion of S+G2/M phase (*Figure 5B*, p<0.0001). Compared with the NC inhibitor group, the percentage of G0/G1 phase cells decreased and the percentage of S+G2/M phase cells increased when *Mir204* and *Mir211* were inhibited (p<0.0001).

Next, we examined the mRNA expression of *Ccnd1*, *Cdkn2a*, *Cdkn1a,* and *Cdkn1c* to confirm the results of flow cytometry. Compared with the cells transfected with *Mir204/Mir211* mimics NC, the expression of *Ccnd1* in the cells transfected with *Mir204/Mir211* mimics was decreased, while expression levels of *Cdkn2a*, *Cdkn1a*, and *Cdkn1c* were increased (*Figure 5C*). Compared with the cells transfected with *Mir204/Mir211* inhibitor NC, *Ccnd1* levels were increased in the cells transfected with *Mir204/Mir211* inhibitors, while expression levels of *Cdkn2a*, *Cdkn1a,* and *Cdkn1c* were significantly decreased. These results revealed that *Mir204/Mir211* affect cell cycle progression through regulation of *Ccnd1*, *Cdkn2a*, *Cdkn1a*, *Cdkn1c* expression.

### *Mir204/Mir211* affect synovial proliferation possibly by regulating PI3K/AKT signaling pathway

To investigate the mechanism of *Mir204/Mir211* on CIA FLS proliferation, we determined changes in PI3K/AKT signaling pathway by western blotting. The results showed that compared with the *Mir204 + Mir211* mimics NC group, the protein levels of PI3K and P-AKT were decreased while P53 expression was increased in the cells transfected with *Mir204/Mir211* mimics (*Figure 5D*). Compared with the cells transfected with *Mir204/Mir211* inhibitor NC, the protein expression of PI3K and P-AKT was up-regulated whereas P53 protein expression was decreased in the cells transfected with *Mir204/Mir211* inhibitors. These results suggest that *Mir204/Mir211* affect synovial proliferation possibly through regulation of PI3K/AKT signaling in CIA FLS.

### *Mir204/Mir211* target *Ssrp1* in CIA FLS

We performed bioinformatic analysis using the databases, TargetScan, miRDB, miRGEN v.3, PicTar, and miRTARbase to predict the downstream target genes of *Mir204/Mir211*. 17 downstream target genes of *Mir204* and 4 downstream target genes of *Mir211* were identified through the cross check with 4 databases (TargetScan, miRDB, miRGEN v.3, and PicTar; *Figure 6A*). Through the second round of cross check with these databases, three same target genes (*Ssrp1*, *M6pr,* and *Sox4*) of *Mir204/Mir211* were identified, and that *Ssrp1* was the most likely downstream target gene of *Mir204/Mir211*.

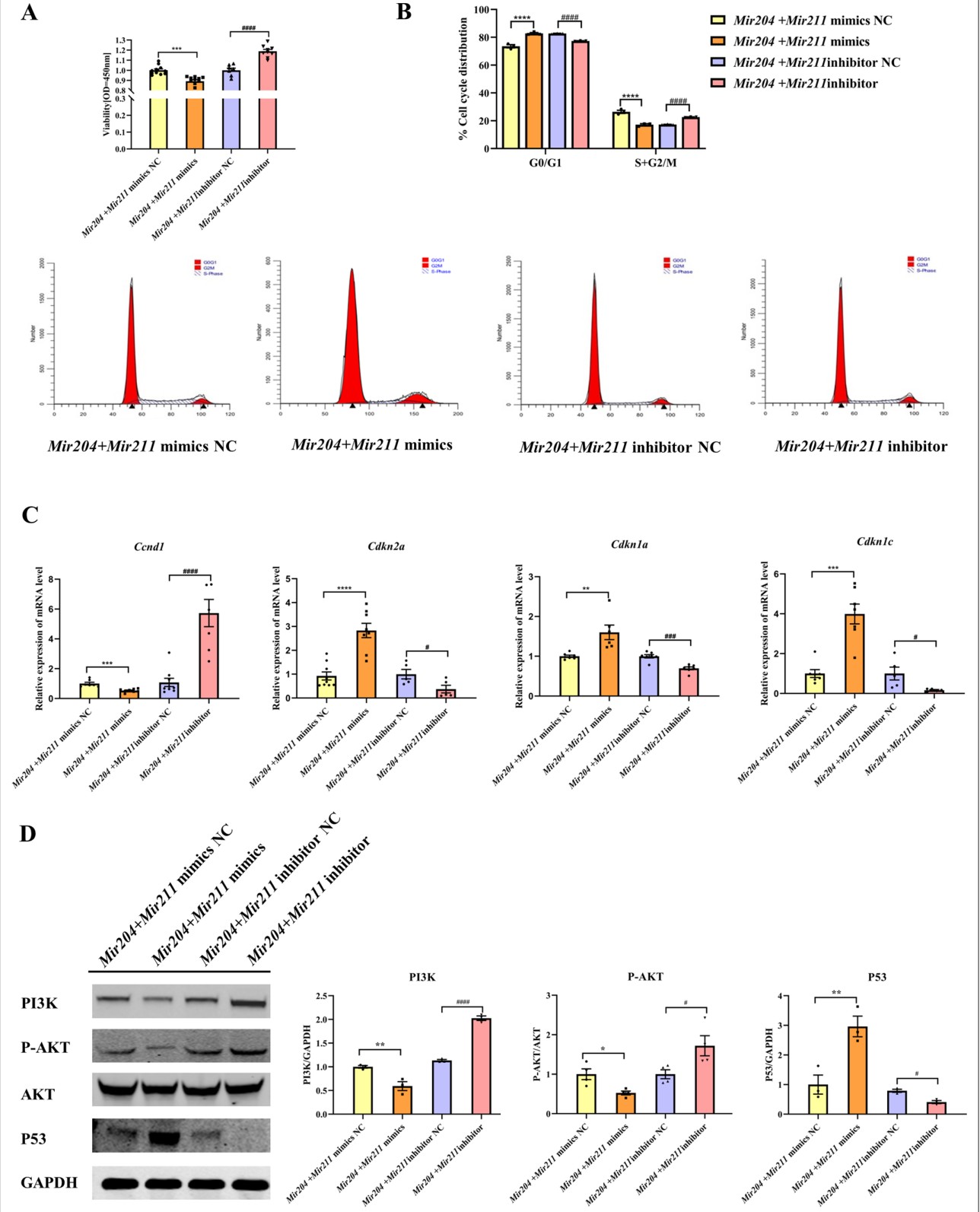

**Figure 5.** Effects of *Mir204/Mir211* on proliferation of collagen-induced arthritis (CIA) fibroblast-like synoviocyte (FLS). (**A**) CCK-8 analysis in four groups of transfected CIA FLS. (**B**) *Mir204/Mir211* induce G0/G1 phase arrest of CIA FLS. (**C**) Expression levels of cell cycle regulatory molecule in transfected CIA FLS. (**D**) Western blot assays of the PI3K/AKT pathway in transfected CIA FLS. Please see *Figure 5—source data 1*. Data are pooled from at

*Figure 5 continued on next page*

*Figure 5 continued*

least three independent experiments and are presented as mean ± SEM. Data are analyzed using one-way ANOVA. $^*p<0.05$, $^{**}p<0.01$, $^{***}p<0.001$, $^{****}p<0.0001$; and $^\#p<0.05$; $^{\#\#}p<0.01$, $^{\#\#\#}p<0.001$, $^{\#\#\#\#}p<0.0001$.

The online version of this article includes the following source data and figure supplement(s) for figure 5:

**Source data 1.** Numerical data obtained during experiments represent in *Figure 5*.

**Source data 2.** Original western blot files for *Figure 5*.

**Figure supplement 1.** Representative images of β-galactosidase staining of primary wild-type (WT) fibroblast-like synoviocyte (FLS) and double knockout (dKO) FLS.

Meanwhile, TargetScan was used to predict the possible *Mir204/Mir211* binding sites in the 3′-UTR of the *Ssrp1* gene (*Figure 6B*).

Expression of SSRP1 protein levels in FLS of CIA rats and CIA mice was significantly higher than those of normal rats and mice (*Figure 6C*), indicating that SSRP1 was highly expressed in CIA. Moreover, compared with the cells transfected with *Mir204/Mir211* mimics NC, the mRNA and protein levels of SSRP1 were significantly decreased in the cells transfected with *Mir204/Mir211* mimics (*Figure 6D*). Compared with the cells transfected with *Mir204/Mir211* inhibitor NC, expression of *Ssrp1* mRNA and protein levels was significantly increased in the cells transfected with *Mir204/Mir211* inhibitors. These results demonstrated that *Ssrp1* may serve as a downstream target of *Mir204/Mir211* in CIA FLS.

## *Mir204/Mir211* affect cell proliferation by targeting *Ssrp1*

As shown in *Figure 7A* and *Figure 7—figure supplement 1*, compared with the control group, the expression of SSRP1 protein in the *Mir204 + Mir211* mimics group was markedly decreased. In comparison with the *Mir204 + Mir211* mimics group, the expression SSRP1 protein in the *Mir204 +Mir211* mimics + SSRP1 group was significantly increased. These results suggest that *Ssrp1* may act as a downstream target of *Mir204/Mir211*.

Effects of *Mir204/Mir211* and SSRP1 on the expression of Ki-67 were examined in 293T cells by IF staining. Compared with the Ctrl group, the number of Ki-67 positive cells was significantly decreased in the *Mir204/Mir211* mimics group; in contrast, the number of Ki-67 positive cells was significantly increased in the *Mir204/Mir211* mimics + SSRP1 group in comparison with the *Mir204/Mir211* mimics group (*Figure 7B*). These results suggest that *Mir204/Mir211* inhibit cell proliferation by regulation of *Ssrp1*.

## High expression level of SSRP1 in CIA *Mir204/Mir211* dKO mice

Compared with WT mice with CIA induction, *Mir204/Mir211* dKO mice with CIA induction showed much more severe paw swelling and increased arthritis score (*Figure 8A*). Compared with CIA WT mice with mild arthritis, CIA *Mir204/Mir211* dKO mice displayed severe arthritis, as manifested by synovial hyperplasia along with angiogenesis and cartilage destruction (*Figure 8B*), suggesting that loss of *Mir204/Mir211* deteriorates RA and *Mir204/Mir211* may play a protective role in RA development. Staining of tartrate resistant acid phosphatase (TRAP) and IHC staining (MMP-13) of knee joints were also performed (*Figure 8—figure supplement 1*).

Results of immunohistochemical (IHC) staining showed that SSRP1 positive cells were significantly increased in synovial tissues of CIA dKO mice compared with those of CIA WT mice (*Figure 8B*). These results suggest that *Ssrp1* may be regulated by *Mir204/Mir211* and is involved in RA development.

## *Ssrp1* knockdown exerts anti-arthritis effect in CIA *Mir204/Mir211* dKO mice

All animal experiments in this part of study were performed in *Mir204/Mir211* dKO mice. Adeno-associated virus (AAV)-*shRNA* Ctrl or AAV-*shSsrp1* was administered to dKO mice with or without CIA induction. The RA phenotype of paw swelling, erythema, joint deformity, and increased arthritis scores was found in CIA dKO mice as compared with the dKO mice without CIA induction (*Figure 9A*). In addition, compared with the CIA dKO mice administered with the AAV-*shRNA* Ctrl, the severity of arthritis phenotype was significantly attenuated in CIA dKO mice administered with AAV-*shSsrp1*. Results of hematoxylin and eosin (H&E), TRAP, and IHC staining of knee joint sections showed that synovial inflammation and articular cartilage destruction were significantly alleviated in CIA dKO mice

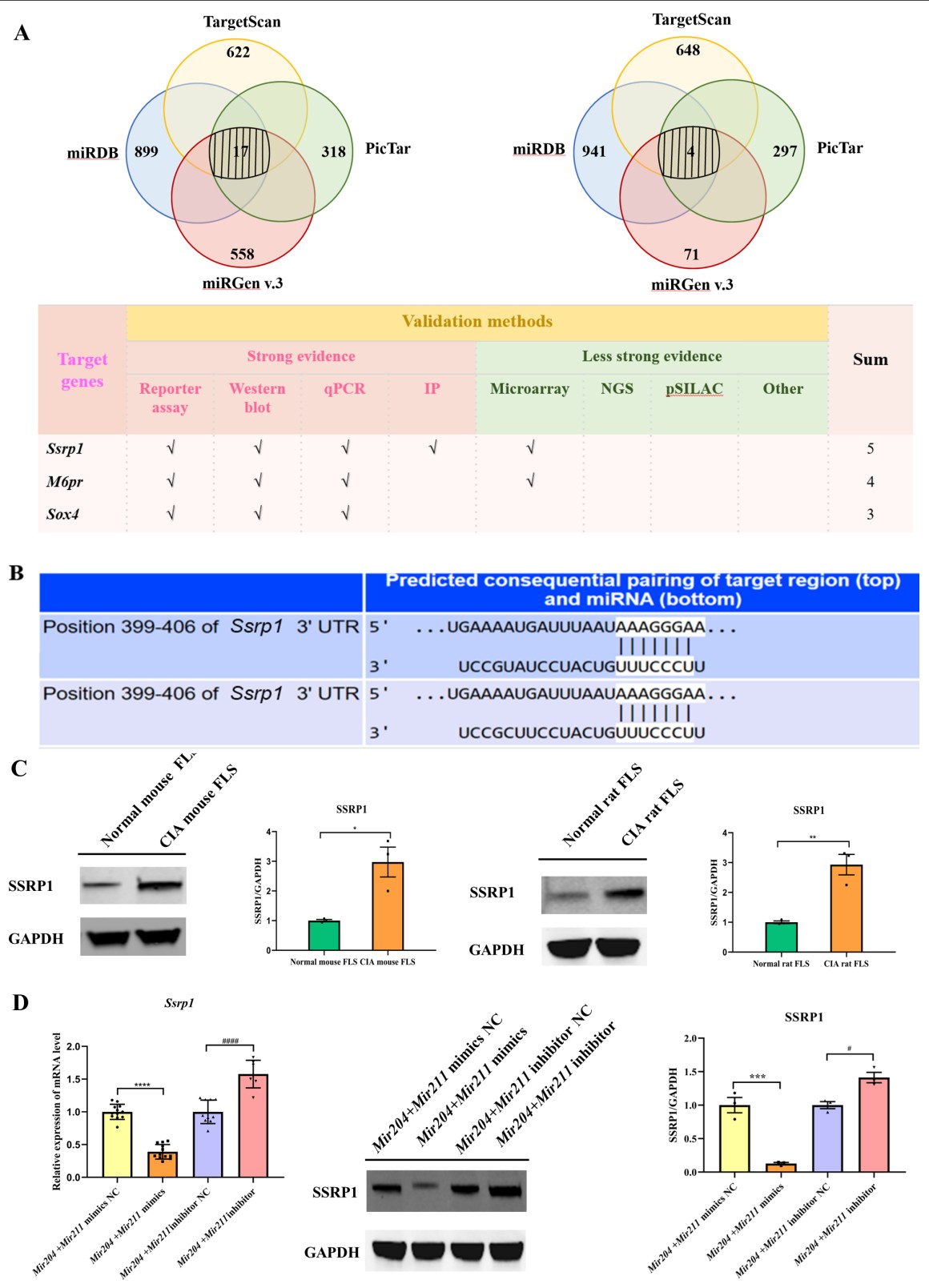

**Figure 6.** *Mir204/Mir211* target structure-specific recognition protein 1 (*Ssrp1*) in collagen-induced arthritis (CIA) fibroblast-like synoviocyte (FLS). (**A**) Diagrams describing the process of predicting target genes of *Mir204/Mir211* (**B**) Diagram of putative *Mir204/Mir211* binding sequence in *Ssrp1* 3' untranslated region (UTR) from TargetScan. (**C**) High expression levels of SSRP1 in CIA mice and CIA rats. (**D**) Gene and protein expressions of SSRP1 in four groups of transfected CIA FLS. Please see *Figure 6—source data 1*. Data are pooled from at least three independent experiments and are

*Figure 6 continued on next page*

*Figure 6 continued*

presented as mean ± SEM. Data are analyzed using unpaired two-tailed Student t test (**C**) and one-way ANOVA (**D**). $^*p<0.05$, $^{**}p<0.01$, $^{***}p<0.001$, $^{****}p<0.0001$; and $^\#p<0.05$; $^{\#\#}p<0.01$, $^{\#\#\#}p<0.001$, $^{\#\#\#\#}p<0.0001$.

The online version of this article includes the following source data for figure 6:

**Source data 1.** Numerical data obtained during experiments represent in *Figure 6*.

**Source data 2.** Original western blot files for *Figure 6*.

administered with AAV-*shSsrp1*, further demonstrating that *Mir204/Mir211* may play an anti-arthritis effect by modulation of *Ssrp1* (*Figure 9B*, *Figure 9—figure supplement 1*).

## Discussion

RA is a chronic inflammatory disease, and genetic and environmental factors are involved during the disease initiation and progression. It is characterized by synovial inflammation leading to abnormal synovial hyperplasia, which eventually invades cartilage and leads to joint destruction (*Li et al., 2018*). The structural similarity and homology between *Mir204* and *Mir211* allow them to have the same gene targeting spectrum (*Lee et al., 2016*). Since the roles of *Mir204* and *Mir211* in RA are still unclear, here we will discuss the key role of *Mir204/211* in RA from two aspects, synovial inflammation and abnormal synovial proliferation. It has been reported that *Mir204* is down-regulated in synovial tissues of RA patients, and *Mir204* regulates RA FLS survival by regulating STAT3 protein (*Xiao et al., 2021*; *Li et al., 2018*). Consistent with these findings, we found that expression levels of *Mir204* and *Mir211* in FLS of CIA mice were significantly reduced compared with those of normal mice. As demonstrated in this study that *Mir204/211* suppress RA progression and play a protective role in RA, in our in vitro cell culture studies, we transfected CIA FLS with *Mir204* mimics (or *Mir204* inhibitor) and *Mir211* mimics (or *Mir211* inhibitor) simultaneously, to verify the important roles of *Mir204/211* in RA.

In in vitro cell culture experiments, simultaneous overexpression of *Mir204/211* was found to alleviate synovial inflammation-associated cell phenotypes in RA. We also performed cell scratching assay and cell apoptosis assay using flow cytometry method. The results suggest that simultaneous overexpression of *Mir204/211* inhibited the migration of CIA FLS and promoted their apoptosis whereas simultaneous knockdown of *Mir204/211* caused opposite phenomena. In RA, inflammatory FLS is the main source of inflammatory cytokines, and these pro-inflammatory mediators will further aggravate synovial inflammation and lead to the RA deterioration (*Wu et al., 2016*; *Wu et al., 2019a*; *Hong et al., 2018*). Yuan et al. reported that the expressions of pro-inflammatory cytokines, such as *Il1b, Il6, and Il23*, were all up-regulated in synovial tissues of CIA mice (*Yuan et al., 2019*). While other cytokines, such as *Il4, Il10,* and *Tgfb1*, played an implicit part in the anti-inflammatory response (*Fabbrini and Magkos, 2015*). In our assays, simultaneous overexpression of *Mir204/211* decreased the expression of inflammatory cytokines (*Il1b, Il6, Il8, and Il23*) in CIA FLS and promoted the synthesis of anti-inflammatory cytokines (*Il4, Il10,* and *Tgfb1*), while knockdown of both *Mir204* and *Mir211* resulted in absolute opposite outcomes. Our results demonstrated that *Mir204/211* treatment suppressed the abnormal migration and apoptotic behavior of RA FLS, as well as inflammatory factor-mediated disease deterioration.

NF-κB is a crucial transcription factors family and acts as a pro-inflammatory mediator to induce the production of cytokines, chemokines, and cell adhesion molecules in inflammatory cells (*Umezawa et al., 2000*; *Imbert and Peyron, 2017*; *Chen et al., 2021*). The translocation of p65 promotes the generation of numerous inflammatory factors (e.g. *Il1b, Il6, Il23*) and matrix metalloproteinases (MMPs) (*Baldwin, 2012*; *Morgan and Liu, 2011*; *Yang et al., 2017*). The expressions of these inflammatory factors and MMPs will eventually lead to synovial inflammation and joint destruction, thereby becoming one of the indispensable pathogenic factors of RA (*Xu et al., 2018*). We found decreased expression levels of P65 and IKKα, increased I-κBα protein expression in the *Mir204 + Mir211* mimics group, while there were promoted P65 and IKKα protein expressions and declined I-κBα expression level in the *Mir204 + Mir211* inhibitor group. Simultaneous overexpression or knockdown of *Mir204/211* had no significant effect on the protein expression of IKKβ, and it seemed *Mir204/211* might regulate the activity of NF-κB signaling pathway through modulation of IKKα. Later, IF assays

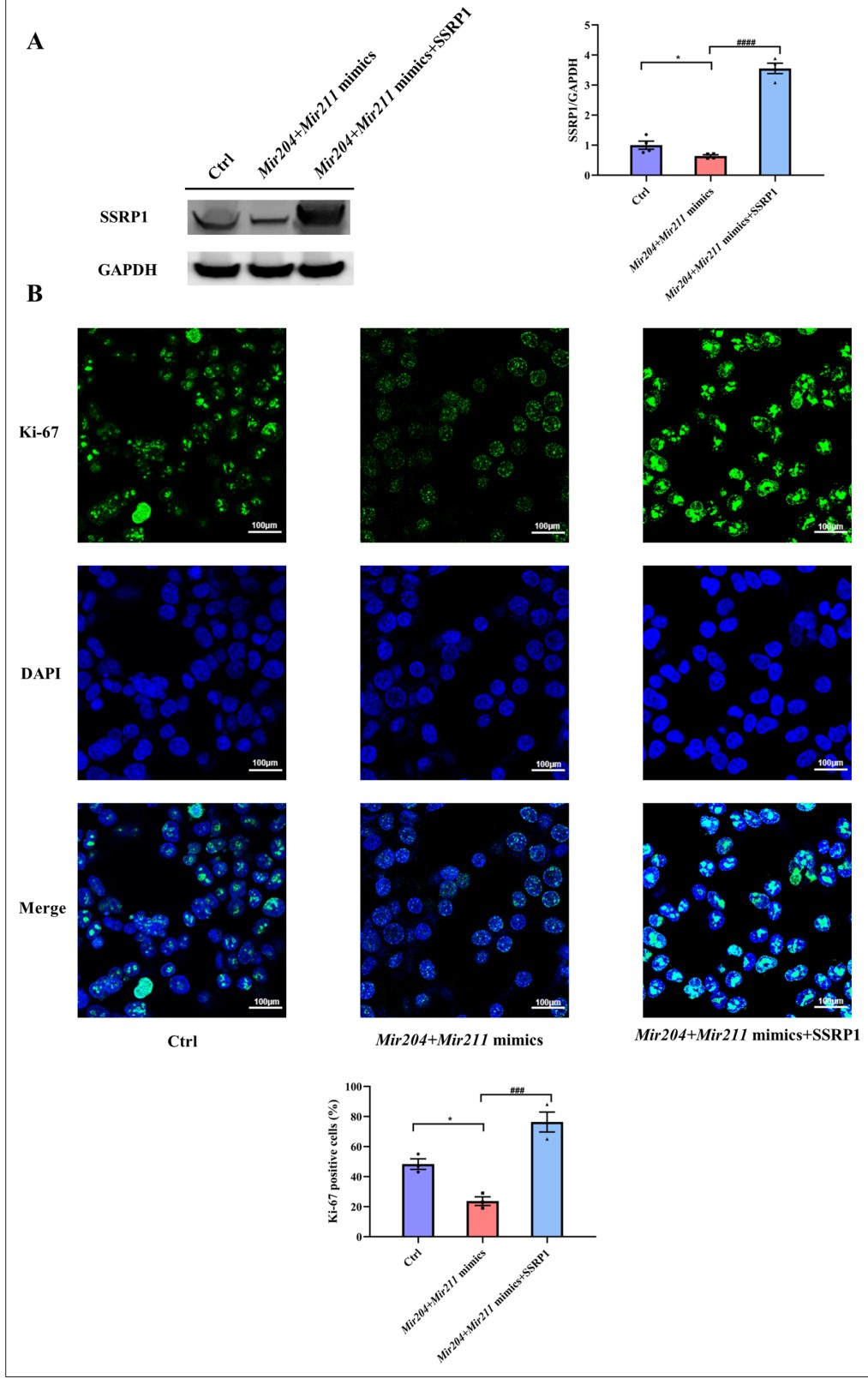

**Figure 7.** Effects of *Mir204/Mir211* on cell proliferation by targeting structure-specific recognition protein 1 (*Ssrp1*). (**A**) Overexpression of *Mir204/Mir211* decreases SSRP1 expression. (**B**) *Mir204/Mir211* decrease Ki-67 levels by targeting *Ssrp1*. Please see *Figure 7—source data 1*. Data are pooled from four independent experiments

*Figure 7 continued on next page*

*Figure 7 continued*

and are presented as mean ± SEM. Data are analyzed using one-way ANOVA. *p<0.05, **p<0.01, ***p<0.001, ****p<0.0001; and #p<0.05, ##p<0.01, ###p<0.001, ####p<0.0001.

The online version of this article includes the following source data and figure supplement(s) for figure 7:

**Source data 1.** Numerical data obtained during experiments represent in *Figure 7*.

**Source data 2.** Original western blot files for *Figure 7*.

**Figure supplement 1.** Overexpression of structure-specific recognition protein 1 (SSRP1) in 293T cells.

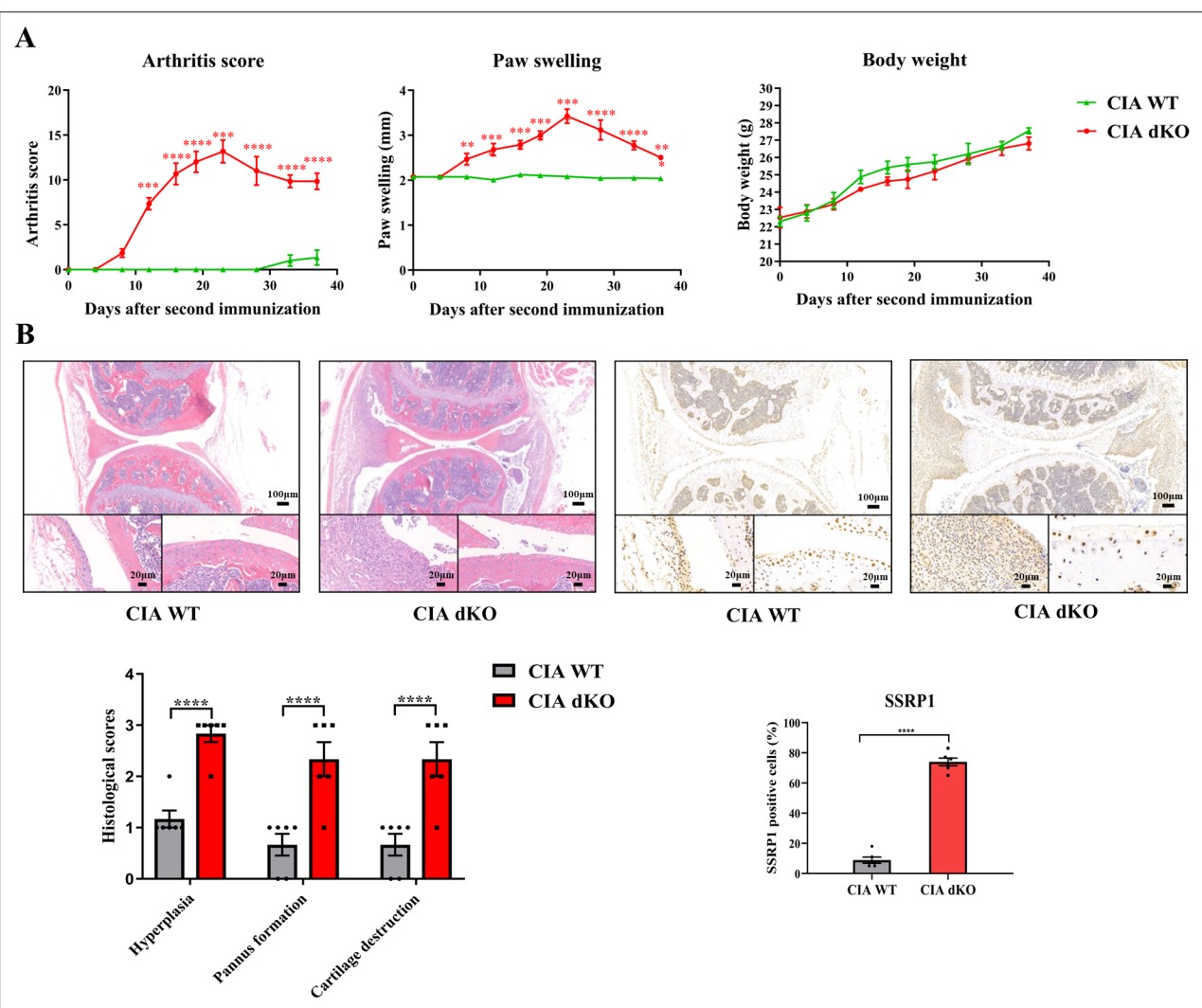

**Figure 8.** High expression level of structure-specific recognition protein 1 (SSRP1) in collagen-induced arthritis (CIA) *Mir204/Mir211* double knockout (dKO) mice. (**A**) Severity of arthritis in CIA wild-type (WT) and CIA dKO mice. The arthritis scores, paw swelling, and body weight are documented by two independent blinded observers since the day of the second booster twice a week (n=6). At 37 days after the second immunization, all mice are sacrificed. Knee joints of all mice are collected for the following experiments. (**B**) Hematoxylin and eosin (H&E) and immunohistochemistry (IHC) analyses of knee joints in CIA WT and CIA dKO mice. Representative photomicrographs of knee joint sections in synovium and cartilage stained with H&E and IHC (SSRP1) are displayed (n=6). Please see *Figure 8—source data 1*. Data are presented as mean ± SEMand analyzed using one-way ANOVA (**A**) and unpaired two-tailed Student t test (**B**). *p<0.05, **p<0.01, ***p<0.001, ****p<0.0001; and #p<0.05; ##p<0.01, ###p<0.001, ####p<0.0001.

The online version of this article includes the following source data and figure supplement(s) for figure 8:

**Source data 1.** Numerical data obtained during experiments represent in *Figure 8*.

**Figure supplement 1.** Representative images of tartrate resistant acid phosphatase (TRAP) and immunohistochemistry (IHC) staining of knee joints in collagen-induced arthritis (CIA) wild-type (WT) and CIA double knockout (dKO) mice.

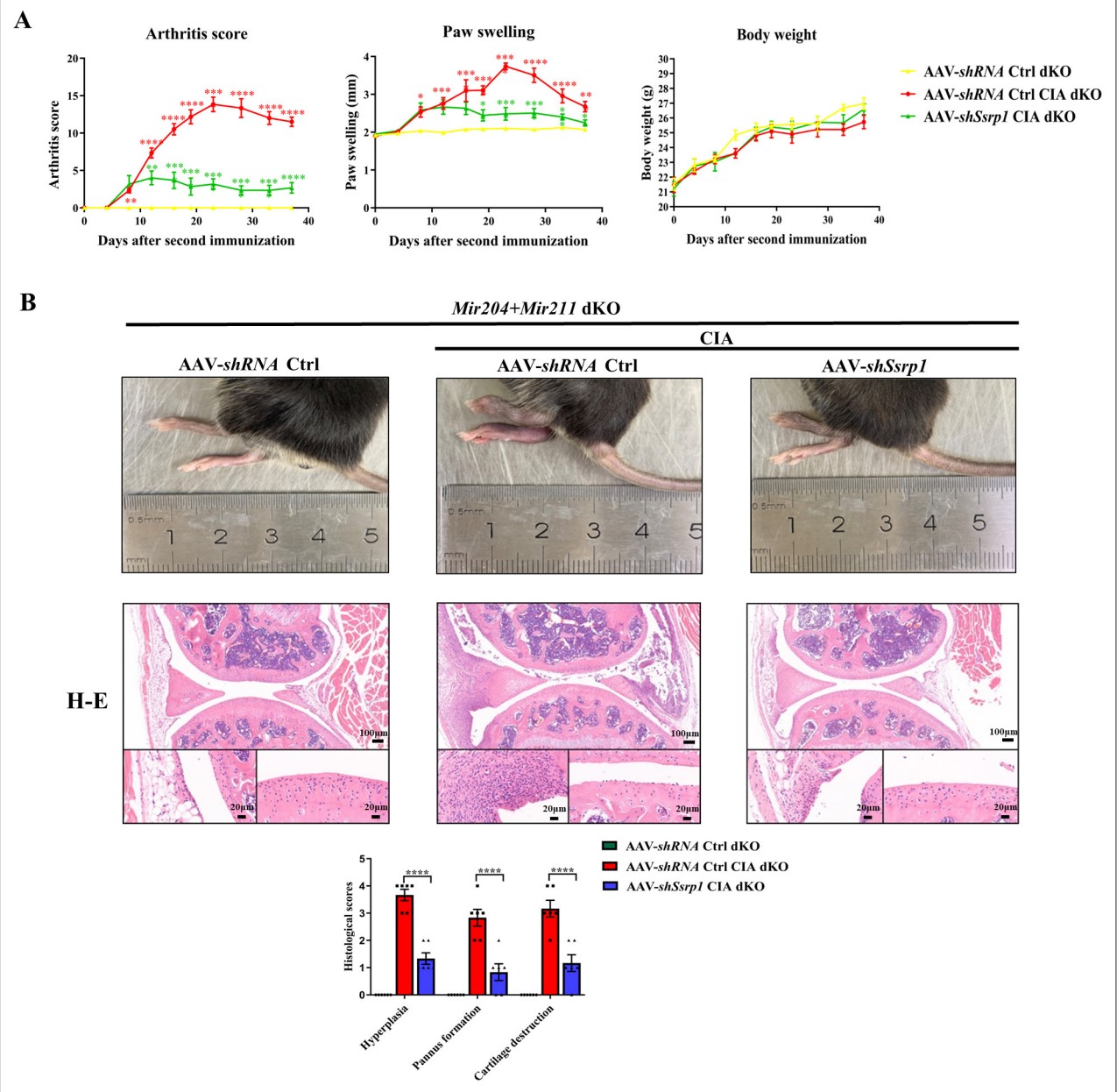

**Figure 9.** Structure-specific recognition protein 1 (*Ssrp1*) knockdown exerts anti-arthritis effect in collagen-induced arthritis (CIA) *Mir204/Mir211* double knockout (dKO) mice. (**A**) *Ssrp1* knockdown ameliorates swelling degrees of hind paws in CIA *Mir204/Mir211* dKO mice. Representative photomicrographs of knee joint sections in synovium and cartilage stained with hematoxylin and eosin (H&E) are displayed (n=6). (**B**) H&E staining of knee joints in *Mir204/Mir211* dKO mice (n=6). Please see *Figure 9—source data 1*. Data are presented as mean ± SEM and are analyzed using one-way ANOVA. *p<0.05, **p<0.01, ***p<0.001, ****p<0.0001; and #p<0.05; ##p<0.01, ###p<0.001, ####p<0.0001.

The online version of this article includes the following source data and figure supplement(s) for figure 9:

**Source data 1.** Numerical data obtained during experiments represent in *Figure 9*.

**Figure supplement 1.** Representative images of tartrate resistant acid phosphatase (TRAP) and immunohistochemistry (IHC) staining of knee joints in three groups of mice.

also confirmed the results of western blotting. Above all, we found that *Mir204/211* attenuated inflammation-mediated exacerbation by regulation of NF-κB p65 translocation in RA.

In terms of cell proliferation, CCK-8 analysis and flow cytometry were both conducted. Cell cycle regulatory proteins include cyclins, cyclin-dependent kinases (CDKs), and CDK inhibitors (CKIs). *Ccnd1* and *Cdkn1c* (CKI) both regulate the cell cycle through regulation of the G1/S phase (*Wang et al.,*

*2020b*). Both *Cdkn2a* and *Cdkn1a* genes are crucial modulators in the regulation of cell senescence and are CDK inhibitors that lead to cell cycle stagnation in the G1 phase (*Fernandez et al., 2015*; *Zhang et al., 2018*). Simultaneous overexpression of *Mir204/211* blocked cell cycle in G0/G1 phase, notably reduced *Ccnd1* expression, and greatly increased *Cdkn2a, Cdkn1a,* and *Cdkn1c* mRNA expression, whereas simultaneous knockdown of *Mir204/211* led to a significant increase in *Ccnd1* expression and a significant decrease in *Cdkn2a, Cdkn1a,* and *Cdkn1c* expressions. *Mir204/211* influenced the cell cycle by regulating *Ccnd1, Cdkn2a, Cdkn1a,* and *Cdkn1c* expression, thus inhibited cell proliferation.

PI3K/AKT is a classical signaling pathway participating in cell proliferation, migration, survival, and angiogenesis. Recent studies have shown that activation of PI3K/AKT signaling promotes the migration of RA FLS and plays an indispensable role in the pathogenesis of RA (*Wang et al., 2020cWang et al., 2020a*; *Jiao et al., 2018*). PI3K regulates cell proliferation, apoptosis, and metabolism. Lin et al. reported increased expression level of PI3K in RA synovial tissue, and the elevated PI3K promoted AKT phosphorylation and activated the PI3K/AKT signaling pathway. Activated AKT phosphorylates a variety of proteins and mediates cell proliferation (*Lin et al., 2019*). Western blotting results revealed that expression of PI3K and P-AKT protein was suppressed in the *Mir204 + Mir211* mimics group, while upregulations of PI3K and P-AKT were observed in the *Mir204 + Mir211* inhibitor group, revealing that *Mir204/211* inhibited aberrant proliferation of CIA FLS by regulating the PI3K/AKT signaling pathway.

Based on the results of bio-informative analysis, PCR assay, and western blot analysis, we found that *Ssrp1* was the downstream target gene of *Mir204/211*. We proposed that the abnormal expression of *Ssrp1* in CIA may be caused by the dysregulation of the upstream miRNAs, *Mir204* and *Mir211*. *Trp53* is a vital tumor suppressor gene, which plays a role in initiating cell death and leads to the inhibition of *Cdkn1a* in cancer cells (*Zhang et al., 2021*). Wu et al. reported that NF-κB and P53 pathways could be regulated by FACT complex, and activation of NF-κB pathway and inhibition of P53 pathway might play the regulatory role of *Ssrp1* in colorectal cancer (*Wu et al., 2019b*). Ding et al. found that in hepatocellular carcinoma, high expression of *Ssrp1* activated the NF-κB pathway and suppressed the P53 pathway. Cyclins, such as *Cdkn1a* and *Cdkn1b,* are downstream signaling factors and can be regulated by *Trp53*. The expression level of *Ccnd1* could also be negatively regulated by *Cdkn1a* (*Vogelstein et al., 2000*). Woksepp et al. demonstrated that knockdown of *Ssrp1* by siRNA enhanced the phosphorylation of P53 and resulted in P53 activation, indicating that *Ssrp1* had an inhibitory effect on *Trp53*. Over-activated NF-κB pathway is a common cause of the inhibition of P53 pathway in tumors, and P53 pathway is negatively regulated by NF-κB pathway (*Gasparian et al., 2011*). Liao et al. reported that low expression of *Ssrp1* depressed p65 expression and a series of proliferation-related genes (such as *Ccnd1*) (*Liao et al., 2017*). Furthermore, Wang et al. also found that *Ssrp1* promoted cell proliferation and apoptosis by activating AKT pathway (*Wang et al., 2019*). Thus, we hypothesized that *Ssrp1* might regulate cell proliferation in RA by cell cycle.

Ki-67 is a widely used proliferation marker, highly expressed in cyclonic cells and significantly downregulated in dormant cells. Ki-67 is an important proliferation marker for various cancer grades (*Sun and Kaufman, 2018*). Ki-67 is well recognized to identify aberrant synovial cells, which display a characteristic of excessive proliferation. Pessler et al. have quantified a strong correlation between Ki-67 expression and the histological degree of synovitis (*Pessler et al., 2008*). Through cell transfection and IF assay, we observed significantly declined expression of Ki-67 in cells with overexpression of *Mir204/211* and prominently promoted Ki-67 expression in cells with overexpression of *Mir204/211* and SSRP1, revealing that *Mir204/211* suppressed cell proliferation by manipulating *Ssrp1* expression.

To further test our hypothesis, we performed the in vivo experiments in the following studies. Severe arthritis, as manifested by synovial hyperplasia along with angiogenesis and cartilage destruction was more significant in mice of the CIA dKO group as compared with mice in the CIA WT group. IHC staining of SSRP1 showed evidently increased SSRP1 positive cells in the knee joint sections of CIA dKO mice as compared with those in CIA WT mice, confirming that *Ssrp1* was the downstream target gene of *Mir204/211,* and the expression of *Ssrp1* was elevated in RA. Lastly, we found that *Ssrp1* knockdown exerts an anti-arthritis effect in CIA dKO mice by regulating proliferation and inflammation. CIA dKO mice received AAV-*shSsrp1* displayed alleviated paw swelling, arthritis scores, synovial inflammation, and articular cartilage destruction compared with the AAV-*shRNA* Ctrl CIA dKO mice. Our study revealed that *Ssrp1*knockdown exerted an anti-arthritis effect in RA by suppressing cell proliferation and inflammation.

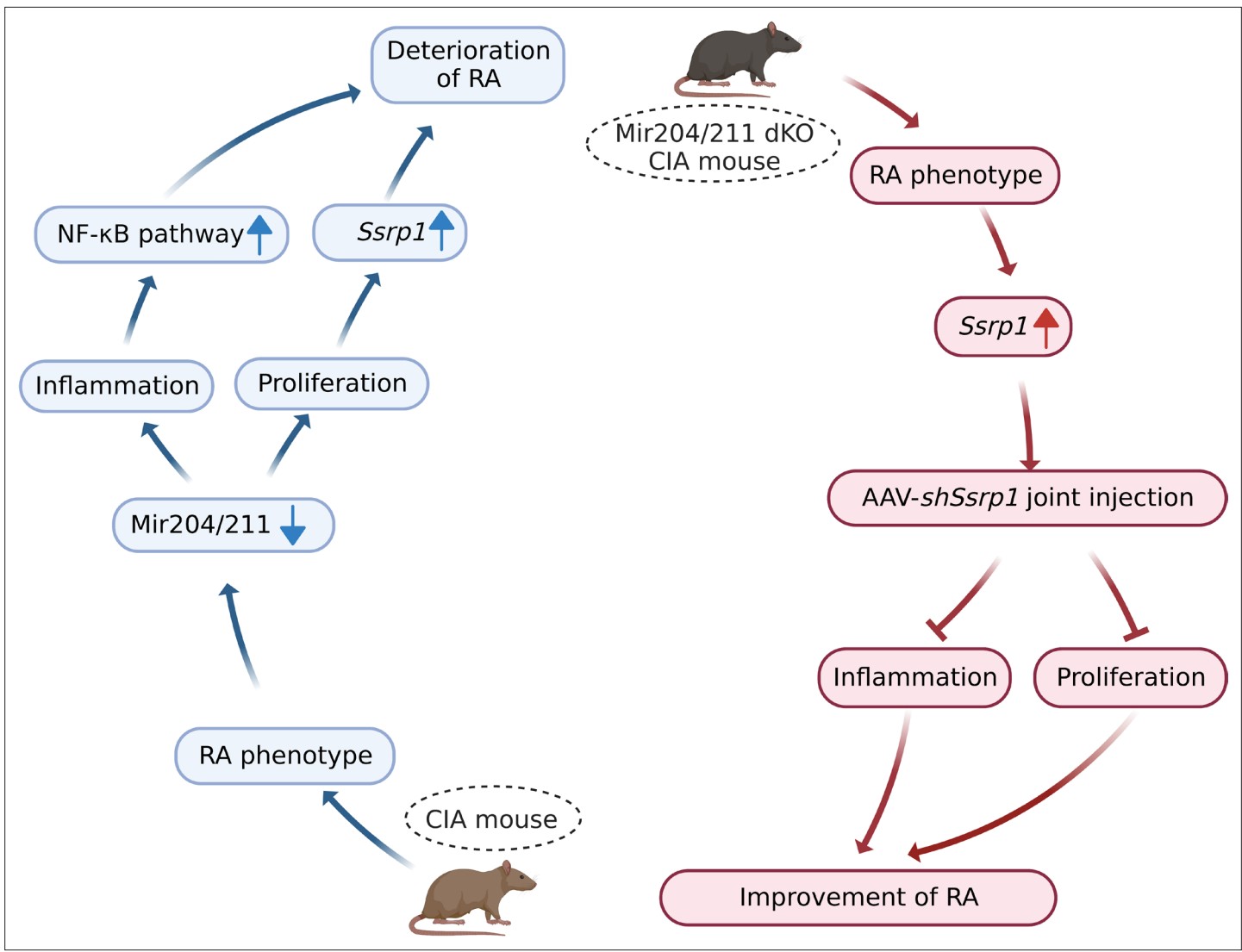

**Figure 10.** Graphical summary of how *Mir204/211* suppresses synovial inflammation and proliferation in rheumatoid arthritis (RA) by targeting structure-specific recognition protein 1 (*Ssrp1*). Primary fibroblast-like synoviocytes (FLSs) derived from collagen-induced arthritis (CIA) mice display decreased expression of *Mir204/211*. On the one hand, upon being challenged by IL-1β, FLS from CIA mice with *Mir204/211* overexpression inhibits the production of inflammatory cytokines and promotes the production of large amounts of anti-inflammatory factors by regulating NF-κB pathway. On the other hand, *Mir204/211* overexpression induces declined cell proliferation by targeting *Ssrp1*. Moreover, *Mir204/211* dKO CIA mice displays RA phenotype and increased SSRP1 expression whereas AAV-*shSsrp1* joint injection significantly reverses this phenomenon from the aspects of inflammation and proliferation. Consequently, we assume that *Mir204/211* may influence cell proliferation and inflammation in RA by regulating downstream target gene *Ssrp1*.

*Mir204/211* suppress RA progression by regulating both inflammation and cell proliferation, and *Mir204/211* affect cell proliferation and inflammation in RA by regulating downstream target gene *Ssrp1* (**Figure 10**).

## Conclusion

RA is a chronic disease characterized by proliferation and infiltration of FLS. *Mir204/211* could delay the progression of RA by regulating both inflammation and cell proliferation. *Ssrp1* is the downstream target gene of *Mir204/211*, and *Mir204/211* affect cell proliferation and retard RA progression by regulating *Ssrp1*. *Ssrp1* plays a critical role in RA development. Further exploration is needed to investigate the interaction between *Mir204/211* and *Ssrp1* in RA.

## Materials and methods
### Animals
Male DBA/1 J mice, male C57 mice, and male Wistar rats (aged 6–8 weeks) were purchased from the Shanghai SLAC Animal Center (Shanghai, China). Germline deletions of the *Mir204/211* by breeding *Mir204flox* and *Mir211flox* mice with CMV-Cre mice were generated and reported in our previous studies (*Huang et al., 2019*). All male mice and rats were placed in a specific pathogen free facility and acclimatized to environment before induction of CIA model. The animal protocol for this study has been approved by the Ethics Committee of the Ninth People's Hospital affiliated to Shanghai Jiao Tong University School of Medicine (ID HKDL[2018]344).

### Induction of CIA model
To establish CIA model in DBA/1 J mice, complete Freund's Adjuvant (CFA, 4 mg/mL) was emulsified with bovine collagen II (1:1). On day –21, DBA/1 J mice were immunized by multiple subcutaneous injections of collagen emulsion around the base of the tail, and a booster of the same dose of emulsion was administrated in the same site to further induce immunization on day 0 (the day of the second booster administration). On day 15, paw swelling and arthritis scores were assessed to confirm whether the CIA mouse model was successfully constructed. CIA rat model was induced with emulsification of bovine collagen II (2 mg/mL) and incomplete Freund's Adjuvant at ratio of 1:1 around the base of the rat tail on day –7 and day 0.

To induce CIA model in *Mir204/211* dKO male mice (C57 background) and their WT littermates, 6.67 mg/mL CFA were emulsified with chicken collagen II (1:1). On day –14, 8-week-old mice were immunized with injections of chicken collagen II emulsion around the tail. On day 0, the same emulsion was injected again in the same place to strengthen immunity and establish CIA model.

### Isolation and culture of FLS
FLSs were isolated from normal and CIA DBA/1 J mice. CIA mice which showed severe paw swelling and an arthritis score of 16 were scarified on day 15 after the second immunization. The individual bones of the hind paws were isolated and dissected. Dissected bones with synovial tissue were digested in Dulbecco Modified Eagle Media (DMEM) containing 1 mg/mL collagenase IV and 0.1 mg/mL dexydiboyydease I for 1 hr. Suspended and filtered through a nylon mesh, synovial cells were washed three times in DMEM medium containing 10% heat inactivated fetal bovine serum and then incubated overnight. The culture media were changed every other day, and FLSs used in this study were at passage 2–6.

### Quantitative qPCR of miRNA
MiRNA from FLS was extracted according to the instructions of miRcute miRNA Isolation Kit (TIANGEN, Beijing) and reversely transcribed with the miRNA first strand cDNA synthesis (Sangon Biotech, Shanghai). The expression levels of *Mir204* and *Mir211* were detected, respectively, by quantitative PCR with the miRNA fluorescence quantitative PCR kit (Sangon Biotech, Shanghai). The expressions of U6 were used as internal reference, and the experiment was repeated at least three times (*Table 1*).

### Cell transfection and intra-articular injection of AAV
NC (Negative control), NC inhibitor, *Mir204* and *Mir211* mimics, *Mir204* and *Mir211* inhibitor, and AAV-mediated *Ssrp1* short hairpin RNA (shRNA) (AAV-*shSsrp1*) and AAV-shRNA Ctrl were synthesized

**Table 1.** PCR primer sequences.

| Gene | Primer sequences |
| --- | --- |
| mmu-*Mir204* forward primer | 5 '- GGGCTTCCCTTTGTCATCCTAT –3' |
| mmu-*Mir211* forward primer | 5 ' - GGGCTTCCCTTTGTCATCCTT –3' |
| Universal U6 Forward primer | 5 ' - GCAAATTCGTGAAGCGTTCCATA –3' |
| Universal PCR reverse primer | 5 ' - AACGAGACGACGACAGAC –3' |

by Genomeditech (Shanghai, China). Empty pcDNA3.1 vector and pcDNA3.1-*Ssrp1* plasmid were purchased from Youze Biotechnology (Changsha, China).

CIA FLSs were seed in six-well plates at the density of 5×10⁵ and randomly divided into *Mir204 + Mir211* mimics NC group, *Mir204 + Mir211* mimics group, *Mir204 + Mir211* inhibitor NC group, and *Mir204 + Mir211* inhibitor group. CIA FLSs were transfected with Lipofectamine 3000 (Invitrogen, America) and 100 nM NC/*Mir204 + Mir211* mimics/NC inhibitor/*Mir204 + Mir211* inhibitor for 48 hr, respectively, according to the protocols. To maintain the inflammatory state and invasive activity of CIA FLS, the medium was discarded and replaced with DMEM containing 10 ng/mL IL-1β for another 24 hr.

293T cells were transfected and divided into control group (transfected with scramble miRNA), *Mir204 + Mir211* mimics group (transfected with *Mir204 + Mir211* mimics), and *Mir204 + Mir211* mimics + SSRP1 group (transfected with *Mir204 + Mir211* mimics and pcDNA3.1-*Ssrp1* plasmid simultaneously). 48 hr after transfection, cells in the three groups were given 10 ng/mL IL-1β for another 24 hr, and the total proteins of the cells were extracted 72 hr after transfection.

Mice in the AVV-*shSsrp1* CIA group underwent joint injections of 1×10¹² AAV-*shSsrp1* particles in a 10 µL volume and the same dose of AAV expressing *shRNA* Ctrl were administered into the knee joints of mice in the AAV-*shRNA* Ctrl and AAV-*shRNA* Ctrl CIA groups.

## Apoptosis analysis and cell migration assay

For Annexin V-APC/PI apoptosis analysis, CIA FLSs in the four groups were transfected, respectively, and collected 72 hr after transfection. After washing twice with PBS, CIA FLSs were resuspended in the binding buffer and stained with Annexin V and propidium iodide (PI) immediately (BD, CA, USA). Cells were collected to detect apoptosis rates.

A straight cell-free scratch was created in transfected CIA FLS with a 20-µL pipette tip. Photographs at specific time points (0, 12, and 24 hr) were documented in four groups to observe the healing of FLS scratches. Wound repair was assessed by ImageJ software (USA).

## Total RNA isolation and qPCR

Total RNA of transfected CIA FLS in four groups was extracted with Trizol reagent (Invitrogen). The cDNA synthesis kit (Takara, Tokyo, Japan) was adopted for reverse transcription of RNA. According to the instructions of the fluorescent quantitative PCR kit (Takara, Japan), real-time PCR was conducted. With *Actb* gene as internal reference, the expression levels of genes in *Mir204 + Mir211* mimics NC group and *Mir204 + Mir211* inhibitor NC group were set as 1, respectively, and the relative expressions of genes in *Mir204 + Mir211* mimics group and *Mir204 + Mir211* inhibitor group were detected

**Table 2.** PCR primer sequences.

| Name | Gene | Forward primer sequence | Reverse primer sequence |
|---|---|---|---|
| β-Actin | *Actb* | 5'-GTGACGTTGACATCCGTAAAGA-3' | 5'-GCCGGACTCATCGTACTCC-3' |
| IL-1β | *Il1b* | 5'-GAAATGCCACCTTTTGACAGTG-3' | 5'-TGGATGCTCTCATCAGGACAG-3' |
| IL-4 | *Il4* | 5'-GGTCTCAACCCCCAGCTAGT-3' | 5'-GCCGATGATCTCTCTCAAGTGAT-3' |
| IL-6 | *Il6* | 5'-CTGCAAGAGACTTCCATCCAG-3' | 5'-AGTGGTATAGACAGGTCTGTTGG-3' |
| IL-8 | *Il8* | 5'-TCGAGACCATTTACTGCAACAG-3' | 5'-CATTGCCGGTGGAAATTCCTT-3' |
| IL-10 | *Il10* | 5'-GCTGGACAACATACTGCTAACC-3' | 5'-ATTTCCGATAAGGCTTGGCAA-3' |
| IL-23 | *Il23* | 5'-CAGCAGCTCTCTCGGAATCTC-3' | 5'-TGGATACGGGGCACATTATTTTT-3' |
| TGF-β1 | *Tgfb1* | 5'-CCACCTGCAAGACCATCGAC-3' | 5'-CTGGCGAGCCTTAGTTTGGAC-3' |
| Cyclin D1 | *Ccnd1* | 5'-GCGTACCCTGACACCAATCTC-3' | 5'-ACTTGAAGTAAGATACGGAGGGC-3' |
| p16 | *Cdkn2a* | 5'-CGCAGGTTCTTGGTCACTGT-3' | 5'-TGTTCACGAAAGCCAGAGCG-3' |
| p21 | *Cdkn1a* | 5'-CCTGGTGATGTCCGACCTG-3' | 5'-CCATGAGCGCATCGCAATC-3' |
| p57 | *Cdkn1c* | 5'-GCAGGACGAGAATCAAGAGCA-3' | 5'-GCTTGGCGAAGAAGTCGTT-3' |
| SSRP1 | *Ssrp1* | 5'-CAGAGACATTGGAGTTCAACGA-3' | 5'-GACGGCTCAATCGAAGCCTC-3' |

**Table 3.** Target proteins detected by western blotting.

| Gene | Protein | Company | Molecular weight (kDa) | Dilution ratio |
|---|---|---|---|---|
| *Rela* | NF-$\kappa$B p65 | Cell Signaling Technology | 65 | 1:1000 |
| *Nfkbia* | I-$\kappa$B$\alpha$ | Cell Signaling Technology | 39 | 1:1000 |
| *Chuk* | IKK$\alpha$ | Cell Signaling Technology | 85 | 1:1000 |
| *Ikbkb* | IKK$\beta$ | Cell Signaling Technology | 87 | 1:1000 |
| *Pik3r1* | PI3K | Cell Signaling Technology | 85 | 1:1000 |
| *Akt1* | P-AKT | Cell Signaling Technology | 60 | 1:1000 |
| *Akt1* | AKT | Cell Signaling Technology | 60 | 1:1000 |
| *Trp53* | P53 | Cell Signaling Technology | 53 | 1:1000 |
| *Ssrp1* | SSRP1 | Cell Signaling Technology | 81 | 1:1000 |
| *Gapdh* | GAPDH | Cell Signaling Technology | 37 | 1:1000 |

(*Table 2*). The experiment was repeated independently for three times, with three replicates for each sample.

## Western blotting

Total proteins were extracted with sodium dodecyl sulfate (SDS) reagent (Beyotime, Shanghai), and eBlot L1 Protein Transfer System (GenScript Corporation, China) was used for protein electrophoresis. SDS-PAGE gel electrophoresis, protein transfer, and immunohybridization were performed to determine changes in expression levels of target genes (*Table 3*). Finally, the target bands were scanned by a fluorescence scanning imager, and the data were quantitatively analyzed.

## Immunofluorescence assay

Translocation of p65 and expression of Ki-67 were confirmed by IF assay. After being fixed in paraformaldehyde for 15 min, cells were permeabilized with 0.3% Triton X-100 for 10 min, blocked with goat serum for 1 hr, and incubated with primary antibody overnight. Next, cells were washed and incubated with corresponding secondary antibody in the dark for 1 hr. 4′,6-Diamidino-2′-phenylindole (DAPI,Beyotime, Shanghai) was adopted to counterstain the nuclei, and the fluorescent signals were immediately detected by laser confocal microscopy.

## CCK-8 assay and flow cytometric assays of cell cycle

At 72 hr after transfection, the media in the four groups were replaced with 1 mL complete medium containing 10% CCK-8 reagent and incubated at 37°C for 4 hr. The absorbance of each well at 450 nm was read using a microplate meter. The experiment was repeated independently for three times, with three replicates for each sample.

The effects of *Mir204/211* on CIA FLS cell cycle were evaluated according to cell cycle detection kit (Beyotime, Shanghai). Transfected cells in four groups were collected 72 hr after transfection. Cell cycle assays were conducted with PI staining reagent in a flow cytometer (BD, CA).

### Prediction of downstream target genes of *Mir204/211*

The TargetScan (http://www.targetscan.org), miRDB (http://miRdb.org/ miRDB/ index.html), miRGEN v.3 (https://www.microrna.gr/mirgen), PicTar (https://pictar.mdc-berlin.de/), and miRTarBase (http://mirtarbase.mbc.nctu.edu.tw/) were used to predict the downstream target genes of miR-204 and miR-211, respectively. Through the intersection of the four databases (The TargetScan, miRDB, miRGEN v.3 and PicTar), the downstream target genes of *Mir204* and *Mir211* were predicted. Through the re-intersection of the predicted downstream target genes of *Mir204* and *Mir211*, the same target genes of *Mir204* and *Mir211* were obtained. The probability of each predicted target gene was verified by the databases miRTarbase and miRGen v.3 again. Meanwhile, TargetScan was used to predict the possible binding sites between target genes and *Mir204/211*.

### Behavioral evaluation, immunohistochemistry, and histological analysis

Body weight, paw swelling, and arthritis scores of each mouse were measured twice a week since day 0. To evaluate the severity of arthritis, the swelling of the forelimbs and hindlimbs of each mouse was assessed using the AI. A score of 0–4 for each paw and 0–16 for each mouse was calculated twice a week, where 0 represents normal and 4 represents severe swelling and joint deformity.

Mice were sacrificed 37 days after the second immunization. Knee joints were retained and fixed in 4% paraformaldehyde for paraffin embedding and pathological observation.

Knee joints of hind legs were decalcified in 10% EDTA and embedded in paraffin. 5-μm thick sections were obtained and underwent H&E staining for histological analysis. The pathological changes were assessed by two blinded observers from aspects of hyperplasia, pannus formation, and cartilage destruction, where 0 indicated normal cell structure without inflammation, and 3 exhibited excessive inflammation with pannus formation and severe articular cartilage damage.

IHC staining was performed on knee joint tissues. Sections were dewaxed in xylene, hydrated with gradient alcohol, incubated overnight with the corresponding primary antibody at 4°C. The primary antibody was then discarded, incubated with the secondary antibody at room temperature for 30 min, and stained with Mayer hematoxylin. The number and percentage of positive-stained cells were observed under the microscope.

### Statistical analysis

All experiment data were expressed as mean ± SEM, and data analyses were conducted with SPSS20.0. One-way ANOVA followed by the Tukey Post-Hoc test was adopted for multiple comparisons of statistical differences, and Students' $t$-test was performed for comparisons between two groups. A p-value less than 0.05 was regarded as statistical significance.

## Acknowledgements

This work was supported by the National Natural Science Foundation of China (NSFC) grants 82172383 and 81874011 to T-YW and NSFC grants 82030067 and 82161160342 to DC. This work was also partially supported by the Shanghai Municipal Science and Technology Commission [Innovation Grant 18140903502 (to T-YW)].

## Additional information

### Funding

| Funder | Grant reference number | Author |
|---|---|---|
| National Natural Science Foundation of China | 82172383 | Ting-Yu Wang |
| National Natural Science Foundation of China | 82030067 | Di Chen |
| National Natural Science Foundation of China | 81874011 | Ting-Yu Wang |
| National Natural Science Foundation of China | 82161160342 | Di Chen |

The funders had no role in study design, data collection and interpretation, or the decision to submit the work for publication.

### Author contributions

Qi-Shan Wang, Kai-Jian Fan, Conceptualization, Data curation, Software, Formal analysis, Investigation, Methodology, Writing - original draft; Hui Teng, Data curation, Software, Investigation, Methodology; Sijia Chen, Data curation, Software, Methodology; Bing-Xin Xu, Data curation, Software, Formal analysis, Methodology; Di Chen, Data curation, Project administration, Writing - review and editing; Ting-Yu Wang, Funding acquisition, Project administration, Writing - review and editing

## Author ORCIDs

Qi-Shan Wang ![ORCID] http://orcid.org/0000-0003-0985-3993
Kai-Jian Fan ![ORCID] http://orcid.org/0000-0002-5295-3175
Di Chen ![ORCID] http://orcid.org/0000-0002-4258-3457
Ting-Yu Wang ![ORCID] http://orcid.org/0000-0001-5000-7890

## Ethics

Animal protocol for this study has been approved by Animal Ethics Committee of Shanghai Ninth People's Hospital (ID HKDL[2018]344).

## Decision letter and Author response

Decision letter https://doi.org/10.7554/eLife.78085.sa1
Author response https://doi.org/10.7554/eLife.78085.sa2

---

# Additional files

## Supplementary files

• Transparent reporting form

## Data availability

All data generated or analysed during this study are included in the manuscript and supporting file.

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
