## [Editor Report]

This important study provides new understanding on the role of miR-204/-211 in the progression of rheumatoid arthritis and the underlying mechanisms. The evidence supporting the conclusions is convincing, with rigorous in vitro cell culture assays and in vivo mouse studies. The work will be of interest for skeletal biologists studying the pathogenesis of rheumatoid arthritis.

---

## [Decision Letter]

**Decision letter after peer review:**

Thank you for submitting your article "MiR-204 and miR-211 suppress synovial inflammation and proliferation in rheumatoid arthritis by targeting SSRP1" for consideration by *eLife*. Your article has been reviewed by 3 peer reviewers, including Mei Wan as the Reviewing Editor and Reviewer #1, and the evaluation has been overseen by a Reviewing Editor and Mone Zaidi as the Senior Editor. The following individual involved in review of your submission has agreed to reveal their identity: Jie Shen (Reviewer #3).

Essential revisions:

1) It is important for the authors to show the distribution of miR-204/211/SSRP1 in joints

2) It is necessary to test the effects of miR-204/-211 dKO on matrix-degrading proteases and joint pain by performing pain tests.

3) The authors should also address the concerns raised in the in vitro studies.

*Reviewer #1 (Recommendations for the authors):*

In the present study, the authors investigated the effects and molecular mechanisms of miR-204/-211 on synovial inflammation and hyperproliferation in rheumatoid arthritis (RA). In the primary fibroblast-like synoviocyte (FLS) culture studies, the authors found that the miR-204/-211 suppresses inflammation and cell proliferation. Taking advantage of a miR-204/-211 dKO mice, the authors demonstrated that miR-204/-211 attenuated synovial inflammation and excessive proliferation in RA model. Moreover, the authors identified the structure-specific recognition protein 1 (SSRP1) as a downstream target gene of miR-204/-211. Importantly, the authors showed that RA phenotypes observed in miR-204/-211 deficient mice were significantly decelerated by intra-articular administration of AAV-shSSRP1. Overall, the study provides useful new information on the role of miR-204/-211 in antagonize synovial inflammation and hyperproliferation during RA progression. The authors should consider the following concerns:

1) The authors should explain in the Introduction section on why miR-204/-211 not other miRNAs were chosen to study in RA development. It would be helpful if the authors give a comprehensive introduction on how miRNAs are involved in the pathogenesis of RA in general.

2) In the in vitro studies, it is unclear why spleen lymphocytes were used. It is also unclear how the effects of MiR-204/-211 on apoptosis and cell migration are relevant to RA progression.

3) The finding that miR-204/miR-211 induces cell cycle arrest and upregulate p16 and p21 gene expression is interesting. Cell cycle arrest with p16 and p21 upregulation is a hallmark change of cellular senescence. It would be helpful if the authors test whether miR-204/miR-211 induces cellular senescence in vitro and in vivo. Cellular senescence may play beneficial role sometimes depending on specific conditions and tissue microenvironments (Int Rev Cell Mol Biol. 2019;346:97-128; Bone Res. 2021 Sep 10;9(1):41). If miR-204/miR-211 indeed induces cellular senescence, it may add another cellular mechanism through which miR-204/miR-211 antagonizes RA progression.

4) In an earlier study, the authors demonstrated that miR-204/-211 loss-of-function induces matrix-degrading proteases in articular chondrocytes and synoviocytes, stimulating articular cartilage destruction. It is unclear whether this is also a mechanism underlying miR-204/-211-regulated RA progression.

5) During RA development, osteoclasts are responsible for bone destruction adjacent to rheumatoid synovium. It is unclear whether miR-204/-211 also affect osteoclast activities.

6) Figure 8A and 8B were not appropriately cited in the text.

7) The images in Figure 8B are not clear. High magnification images are needed.

8) There are instances of poor English usage that need to be addressed by appropriate editing.

*Reviewer #2 (Recommendations for the authors):*

RA is a chronic joint disease and needs more effective therapies. The current study used a mouse model of RA and indicated that miR-204/-211 and SSRP1 are new molecules that involve in RA development, and serves as new therapeutic targets for RA. Data show (1) decreased expression of miR-204/-211 in CIA joints and FLS. Overexpression or knockdown of miR-204/-211 in CIA FLS reduces or increased growth, migration, and inflammatory cytokines, respectively. Bioinformatic analysis reveals that SSRP1 is the downstream target gene of miR-204/-211. Further, 204/-211dKO/CIA mice have more severe RA, which is attenuated by SSRP1shRNA. Conclusion is that miR-204/-211 and SSRP1 play a critical role in RA development and treatment. Further exploration is needed to investigate the interaction between miR-204/-211 and SSRP1 in RA. The study has many strengths as follows. (1) The role of miR-204/-211/SSRP1 in RA pathogenesis has not been studied. (2) Decreased expression of miR-204/-211 in CIA FLS is new and significant. (3) Both gain and loss of function approach are used. (4) Provided in vivo evidence to demonstrate the involvement of miR-204/-211/SSRP1.

1) Change all figures to dot-plot format.

2) CIA model, the arthritis incidence for mice with DBA background and mir-204/211 KO mice should be provided, specifically since it is difficult to induce CIA arthritis in B6 background. Sex of mir-204/211 KO mice should be provided.

3) Day of CIA when FLS is isolated and joint inflammation status at this time should be provided.

4) FLS is an important cell type for RA pathogenesis. However, other cell types such as macrophages and B and T lymphocytes also play a role in RA. Thus it will be important to examine the distribution of miR-204/-211/SSRP1 in joints by IHC or flow cytometry to demonstrate that fibroblasts are the main cells that express miR-204/-211/SSRP1. Another suggestion is to perform bioinformatic analysis using published scRNAdata from mouse or human RA synovium to show fibroblasts are the main cells that express miR-204/-211/SSRP1. These data will provide a strong rationale for focusing FLS in the current study.

5) The effect of mir204/211 mimics in CIA will be a critical experiment. Another point is if the focus is on mir204/211 target SSRP1, SSRP1 shRNA should be tested in CIA mice to see if SSRP inhibition should attenuate joint lesion for clinical relevance. The rationale of using mir204/211KO CIA mice is not clearly described.

6) It is not clear why CIA FLS is used in all in vitro experiments. Fundamental finding of this study is that CIA FLS has reduced mir204/211 expression, leading to FLS proliferation-inflammation. If this is the case, WT FLS should be used to see if mir204/211 inhibitor could make WT FLS expressing CIA FLS phenotype. Thus, it makes more sense to use WT FLS in loos of function experiment (mir204/211 inhibitor experiments). However, the overexpression experiment using CIA FLS is OK to show if over-expression in CIA FLS could rescue the phenotypes caused by low mir204/211 expression.

7) RA is an autoimmune disease and multiple organs may be involved. However, the rationale of using spleen cells in this study is not clear. Why spleen cells have reduced mir204/211 expression in CIA mice if the inflammation occurs in the joints? Is this a systemic effect? If the authors want to demonstrate the systemic changes of mir204/211 in RA, examination of mir204/211 in blood cells such as lymphocytes is more meaningful. I suggest eliminating the spleen data unless the authors provide a strong rationale.

*Reviewer #3 (Recommendations for the authors):*

In the manuscript, the authors are trying to investigate the effect of miR-204/miR-211 on synovial inflammation and hyperplasia during the progression of rheumatoid arthritis (RA), and further explore the mechanisms involved in the regulation of miR-204/miR-211 for synovial homeostasis. The authors provided a large amount of data to demonstrate that miR-204/miR-211 could mitigate synovial inflammation and attenuate excessive proliferation of fibroblast-like synoviocytes in RA. Moreover, they showed that SSRP1 could serve as the downstream target gene of miR-204/-211, and intra-articular administration of AAV-shSSRP1 could alleviate synovial inflammation and articular cartilage destruction in CIA miR-204/-211 dKO mice.

Overall, the study is well-designed and well-conducted. The authors conducted in vitro and in vivo studies, and further implement genetic mouse models and CIA-induced RA animal models to demonstrate that miR204/211 could regulate synovial inflammation and proliferation in RA by targeting SSRP1, and thus they could be novel therapeutic targets for RA. It would be better to include a pain behavior test in addition to paw swelling and arthritis scores, since joint pain is the manifestation of RA in clinic.

1) Pain test needs to be included to evaluate the effect of miR204/-211 dKO on RA induced joint pain.

---

## [Author Response]

Reviewer #1 (Recommendations for the authors):In the present study, the authors investigated the effects and molecular mechanisms of miR-204/-211 on synovial inflammation and hyperproliferation in rheumatoid arthritis (RA). In the primary fibroblast-like synoviocyte (FLS) culture studies, the authors found that the miR-204/-211 suppresses inflammation and cell proliferation. Taking advantage of a miR-204/-211 dKO mice, the authors demonstrated that miR-204/-211 attenuated synovial inflammation and excessive proliferation in RA model. Moreover, the authors identified the structure-specific recognition protein 1 (SSRP1) as a downstream target gene of miR-204/-211. Importantly, the authors showed that RA phenotypes observed in miR-204/-211 deficient mice were significantly decelerated by intra-articular administration of AAV-shSSRP1. Overall, the study provides useful new information on the role of miR-204/-211 in antagonize synovial inflammation and hyperproliferation during RA progression. The authors should consider the following concerns:1) The authors should explain in the Introduction section on why miR-204/-211 not other miRNAs were chosen to study in RA development. It would be helpful if the authors give a comprehensive introduction on how miRNAs are involved in the pathogenesis of RA in general.

We have made proper changes in the Introduction section in Line 99-103, 115-119 in the revised manuscript as the reviewer suggested.

2) In the in vitro studies, it is unclear why spleen lymphocytes were used. It is also unclear how the effects of MiR-204/-211 on apoptosis and cell migration are relevant to RA progression.

RA is a systemic inflammatory disease with significant changes in immune system, and the spleen is an important immune organ. Changes in spleen function and joint damage have been the focus of RA research. In previous studies, we found marked changes in cell proliferation in splenic lymphocytes and in a subset of T cells during RA progression [1-3]. Functional changes in splenocytes could reflect the immune status and disease conditions of RA. The finding of decreased miR-204/211 expression in spleen cells in CIA mice suggests that miR-204/211 have a significant effect on spleen cells in RA disease. We have eliminated the spleen data according to the suggestion of Reviewer 2.

Synovial inflammation and hyperplasia is suggested to be the major etiopathogenesis of RA. The hyperplasic synovial cells surprisingly display RA-like phenotype, such as anti-apoptosis and aggressive cell migration potential [4,5]. The migration of synovial cells lead to the direct contact of synovial cells with articular cartilage and regulation of articular chondrocyte metabolism. Apoptosis and migration of synovial cells are closely related to the progression of RA. Therefore, we studied how the effects of miR-204/-211 on apoptosis and cell migration.

3) The finding that miR-204/miR-211 induces cell cycle arrest and upregulate p16 and p21 gene expression is interesting. Cell cycle arrest with p16 and p21 upregulation is a hallmark change of cellular senescence. It would be helpful if the authors test whether miR-204/miR-211 induces cellular senescence in vitro and in vivo. Cellular senescence may play beneficial role sometimes depending on specific conditions and tissue microenvironments (Int Rev Cell Mol Biol. 2019;346:97-128; Bone Res. 2021 Sep 10;9(1):41). If miR-204/miR-211 indeed induces cellular senescence, it may add another cellular mechanism through which miR-204/miR-211 antagonizes RA progression.

During cell senescence, senescence-associated β-galactosidase activity is upregulated. Two main signaling pathways mediating the growth arrest of the senescent cells are the ATM/p53/p21Waf1 and the p16INK4a/pRB signaling cascades. The activation of p53 transcription factor increases expression of p21 to arrest the cell cycle. p16INK4a/pRB signaling initiates and maintains permanent cell cycle arrest [6,7]. In our study, we found that miR-204/-211 induce cell cycle arrest and upregulate p16 and p21 gene expression. In the revision, we isolated FLS from WT and miR-204/-211dKO mice and performed a β-galactosidase assay to determine whether miR-204/-211 induce cellular senescence. We found that β-galactosidase activity was down-regulated in miR-204/-211 dKO FLS as compared with WT FLS (Figure 5-supplement 1).

4) In an earlier study, the authors demonstrated that miR-204/-211 loss-of-function induces matrix-degrading proteases in articular chondrocytes and synoviocytes, stimulating articular cartilage destruction. It is unclear whether this is also a mechanism underlying miR-204/-211-regulated RA progression.

As the reviewer suggested, we have performed IHC staining of MMP-13 in the sections of knee joints to determine whether matrix-degrading proteases in articular joint were altered. Compared with CIA WT mice, MMP-13 positive cells were significantly increased in articular joint of CIA dKO mice (Figure 8—figure supplement 1). Furthermore, administration with AAV-shSSRP1 significantly decreased numbers of MMP-13 positive cells as compared with AAV-shRNA Ctrl in CIA dKO mice (Figure 9—figure supplement 1). According to these results, miR-204/-211 loss-of-function induces matrix-degrading proteases in articular joint, stimulating articular destruction in RA.

5) During RA development, osteoclasts are responsible for bone destruction adjacent to rheumatoid synovium. It is unclear whether miR-204/-211 also affect osteoclast activities.

As the reviewer suggested, we have performed TRAP staining to determine whether miR-204/-211 also affect osteoclast activities. Compared with CIA WT mice, generation of osteoclasts were significantly increased in CIA dKO mice (Figure 8—figure supplement 1). Furthermore, administration with AAV-shSSRP1 significantly decreased osteoclast formation as compared with AAV-shRNA Ctrl in CIA dKO mice (Figure 9—figure supplement 1). These results demonstrated that loss-of-function of miR-204/-211 also affects osteoclast formation, which is responsible for bone destruction adjacent to the rheumatic synovium during RA development.

6) Figure 8A and 8B were not appropriately cited in the text.

Proper changes were made in Line 382-384, 389-390 in the revised manuscript as the reviewer suggested.

7) The images in Figure 8B are not clear. High magnification images are needed.

We have provided higher magnification images of Figure 8B in the revised manuscript as the reviewer suggested.

8) There are instances of poor English usage that need to be addressed by appropriate editing.

We have carefully revised the entire manuscript as the reviewer suggested.

Reviewer #2 (Recommendations for the authors):RA is a chronic joint disease and needs more effective therapies. The current study used a mouse model of RA and indicated that miR-204/-211 and SSRP1 are new molecules that involve in RA development, and serves as new therapeutic targets for RA. Data show (1) decreased expression of miR-204/-211 in CIA joints and FLS. Overexpression or knockdown of miR-204/-211 in CIA FLS reduces or increased growth, migration, and inflammatory cytokines, respectively. Bioinformatic analysis reveals that SSRP1 is the downstream target gene of miR-204/-211. Further, 204/-211dKO/CIA mice have more severe RA, which is attenuated by SSRP1shRNA. Conclusion is that miR-204/-211 and SSRP1 play a critical role in RA development and treatment. Further exploration is needed to investigate the interaction between miR-204/-211 and SSRP1 in RA. The study has many strengths as follows. (1) The role of miR-204/-211/SSRP1 in RA pathogenesis has not been studied. (2) Decreased expression of miR-204/-211 in CIA FLS is new and significant. (3) Both gain and loss of function approach are used. (4) Provided in vivo evidence to demonstrate the involvement of miR-204/-211/SSRP1.1) Change all figures to dot-plot format.

We have changed all bar graphs to the dot-plot format as the reviewer suggested.

2) CIA model, the arthritis incidence for mice with DBA background and mir-204/211 KO mice should be provided, specifically since it is difficult to induce CIA arthritis in B6 background. Sex of mir-204/211 KO mice should be provided.

In the previous studies, we found that the successful rate of CIA model in the DBA background strain was 100% [8]. In this study, the successful rate of CIA model in the miR-204/211 KO mice (B6 strain) was over 60%, which was much higher than that in wide-type B6 mice (20%). Furthermore, compared with CIA wide-type B6 mice, miR-204/-211 dKO mice induced with CIA showed much more severe paw swelling and increased arthritis score. Male miR-204/211 KO mice were used in the study and the sex of miR-204/211 KO mice were provided in Line 591, 595, 612.

3) Day of CIA when FLS is isolated and joint inflammation status at this time should be provided.

We have provided the day of CIA when FLS was isolated and joint inflammation status at this time (Line 619-621) in the revised manuscript.

4) FLS is an important cell type for RA pathogenesis. However, other cell types such as macrophages and B and T lymphocytes also play a role in RA. Thus it will be important to examine the distribution of miR-204/-211/SSRP1 in joints by IHC or flow cytometry to demonstrate that fibroblasts are the main cells that express miR-204/-211/SSRP1. Another suggestion is to perform bioinformatic analysis using published scRNAdata from mouse or human RA synovium to show fibroblasts are the main cells that express miR-204/-211/SSRP1. These data will provide a strong rationale for focusing FLS in the current study.

We have examined the distribution of miR-204/-211/SSRP1 in joints as the reviewer suggested. Representative images of Fisher staining of miR-204/-211 in the knee joint sections of WT mice and CIA mice were shown in the revised manuscript. It is clear that miR-204/-211 were mainly expressed in the synovium of knee joint. Compared with WT mice, the expression of miR-204/-211 was lower in the synovium of CIA mice. SSRP1 was also highly expressed in the synovium of knee joint (Figure 6C and 8B). As compared with WT mice, SSRP1 expression was significantly increased in FLS and synovium in CIA mice (Figure 6C and 8B). We thus focused our work on the role of miR-204/-211/SSRP1 in RA FLS in the current study.

**Author response image 1. sa2fig1:** 

5) The effect of mir204/211 mimics in CIA will be a critical experiment. Another point is if the focus is on mir204/211 target SSRP1, SSRP1 shRNA should be tested in CIA mice to see if SSRP inhibition should attenuate joint lesion for clinical relevance. The rationale of using mir204/211KO CIA mice is not clearly described.

Instead of using miR-204/211 mimics in CIA mouse, we used miR-204/211 dKO mice to investigate the role of miR-204/211 in RA progression. Consistent with our in vitro results, we found that miR-204/-211 dKO mice with CIA induction displayed more severe arthritis phenotype as compared with WT mice with CIA induction, suggesting that loss of function of miR-204/-211 aggravates RA progression (Figure 8). To further explore whether miR-204/211 target SSRP1, we performed in vivo experiments and found that RA phenotype observed in miR-204/-211 deficient mice were significantly decelerated by intra-articular administration of AAV-shSSRP1, which was consistent with in vitro results (Figure 9). These in vivo findings confirmed the role of miR-204/-211-SSRP1 signaling during RA development.

6) It is not clear why CIA FLS is used in all in vitro experiments. Fundamental finding of this study is that CIA FLS has reduced mir204/211 expression, leading to FLS proliferation-inflammation. If this is the case, WT FLS should be used to see if mir204/211 inhibitor could make WT FLS expressing CIA FLS phenotype. Thus, it makes more sense to use WT FLS in loos of function experiment (mir204/211 inhibitor experiments). However, the overexpression experiment using CIA FLS is OK to show if over-expression in CIA FLS could rescue the phenotypes caused by low mir204/211 expression.

RA is a multifactorial immune disease. The evidence, obtained from genetic analysis, animal models, and clinical studies, points RA to an immune-mediated disease associated with stromal tissue dysregulation, together propagated chronic inflammation and articular cartilage destruction [9,10]. The phenotype of CIA FLS is completely different from WT FLS. According to the current research, WT FLS switches to the CIA FLS phenotype basically through immune induction which is a complicated process. WT FLS could not switch to CIA FLS in vitro by single miRNA stimulation or modification. In this study, we investigated the importance of miR-204/211 in RA disease progression. Therefore, CIA FLS, but not WT FLS, was used in all in vitro experiments. In in vitro experiments, CIA FLS was stimulated by IL-1β to maintain RA-like disease state so as not to lose the RA phenotype during the research period. We validated the importance of miR-204/211 in RA by lose-of-function and gain-of-function of miR-204/211 in CIA FLS.

7) RA is an autoimmune disease and multiple organs may be involved. However, the rationale of using spleen cells in this study is not clear. Why spleen cells have reduced mir204/211 expression in CIA mice if the inflammation occurs in the joints? Is this a systemic effect? If the authors want to demonstrate the systemic changes of mir204/211 in RA, examination of mir204/211 in blood cells such as lymphocytes is more meaningful. I suggest eliminating the spleen data unless the authors provide a strong rationale.

We have eliminated the spleen data as the reviewer suggested.

Reviewer #3 (Recommendations for the authors):In the manuscript, the authors are trying to investigate the effect of miR-204/miR-211 on synovial inflammation and hyperplasia during the progression of rheumatoid arthritis (RA), and further explore the mechanisms involved in the regulation of miR-204/miR-211 for synovial homeostasis. The authors provided a large amount of data to demonstrate that miR-204/miR-211 could mitigate synovial inflammation and attenuate excessive proliferation of fibroblast-like synoviocytes in RA. Moreover, they showed that SSRP1 could serve as the downstream target gene of miR-204/-211, and intra-articular administration of AAV-shSSRP1 could alleviate synovial inflammation and articular cartilage destruction in CIA miR-204/-211 dKO mice.Overall, the study is well-designed and well-conducted. The authors conducted in vitro and in vivo studies, and further implement genetic mouse models and CIA-induced RA animal models to demonstrate that miR204/211 could regulate synovial inflammation and proliferation in RA by targeting SSRP1, and thus they could be novel therapeutic targets for RA. It would be better to include a pain behavior test in addition to paw swelling and arthritis scores, since joint pain is the manifestation of RA in clinic.1) Pain test needs to be included to evaluate the effect of miR204/-211 dKO on RA induced joint pain.

In our previous work, we have demonstrated that the miR204/-211 dKO mice had increased pain sensitivity tested by behavioral pain tests (von Frey test) and deficiency of miR-204/-211 in mesenchymal progenitor cells and mesenchymal joint tissues contributed to the genesis of OA pain. OA mice receiving miR-204 injection showed higher pain thresholds than those receiving control AAV5 injection, suggesting that miR-204 reintroduction in OA knee joints not only ameliorated histological features, including cartilage degradation and osteophyte formation, but also reduced pain [11]. Since RA is an autoimmune disease and synovial inflammation is the major etiopathogenesis of RA, thereby, in our present work, we concentrated our work on the role of miR-204/-211 in RA synovial inflammation. Since joint pain might also be an indispensable part in RA progression as suggested, we are planning to start a new subject focusing on the role of miR-204/-211 in RA joint pain in the near future.